# Ultrabright gap-enhanced Raman tags for high-speed bioimaging

Yuqing Zhang[1,2], Yuqing Gu[1], Jing He[1], Benjamin D. Thackray[1] & Jian Ye [1,3,4]

Surface-enhanced Raman spectroscopy (SERS) is advantageous over fluorescence for bioimaging due to ultra-narrow linewidth of the fingerprint spectrum and weak photo-bleaching effect. However, the existing SERS imaging speed lags far behind practical needs, mainly limited by Raman signals of SERS nanoprobes. In this work, we report ultrabright gap-enhanced Raman tags (GERTs) with strong electromagnetic hot spots from interior sub-nanometer gaps and external petal-like shell structures, larger immobilization surface area, and Raman cross section of reporter molecules. These GERTs reach a Raman enhancement factor beyond $5 \times 10^9$ and a detection sensitivity down to a single-nanoparticle level. We use a 370 μW laser to realize high-resolution cell imaging within 6 s and high-contrast (a signal-to-background ratio of 80) wide-area ($3.2 \times 2.8$ cm$^2$) sentinel lymph node imaging within 52 s. These nanoprobes offer a potential solution to overcome the current bottleneck in the field of SERS-based bioimaging.

[1] State Key Laboratory of Oncogenes and Related Genes, School of Biomedical Engineering, Shanghai Jiao Tong University, Shanghai 200030, P. R. China. [2] School of Automation, Hangzhou Dianzi University, Hangzhou 310018, P. R. China. [3] Shanghai Key Laboratory of Gynecologic Oncology, Ren Ji Hospital, School of Medicine, Shanghai Jiao Tong University, Shanghai 200127, China. [4] Department of Nuclear Medicine, Ruijin Hospital, School of Medicine, Shanghai Jiao Tong University, Shanghai 200025, China. Correspondence and requests for materials should be addressed to J.Y. (email: yejian78@sjtu.edu.cn)

Surface-enhanced Raman scattering (SERS) technique is of crucial scientific and practical interest in many fields, such as chemistry, physics, biology, and medicine[1–7]. SERS tags, as one of the most popular optical labels[8–10], have exhibited great potential for bioimaging with advantages of the unique fingerprint vibrational spectrum and the ultra-narrow spectral linewidth. While fluorescent tags can be bright, they typically exhibit poor photostability, allow limited multiplexing due to their broad and overlapped fluorescence spectra, and suffer the autofluoscence from biological tissues in in vivo applications. SERS tags have shown great prospects for in vivo bioimaging applications such as bright optical guidance for intraopeative detection of cancer margins, residual cancerous foci, microscopic or disseminated tumors, and sentinel or metastasized lymph node[6,11–18]. However, the existing SERS imaging speed lags far behind clinical needs. For example, it typically takes hours to acquire a wide-area Raman in vivo image[13,16]. The current bottlenecks for high-speed SERS bioimaging mainly include the overall Raman signals (instead of the Raman enhancement factor) of SERS tags and imaging method of Raman system.

There have been a lot of efforts to design and synthesize SERS tags that can generate bright and stable SERS signals[19,20]. The intensity of SERS tags largely rely on the creation of electromagnetic hot spots, i.e., spatially localized regions with extremely strong electromagnetic fields[21]. In addition to the creation of sharp tips and rough surfaces through morphologically controlled synthesis, hot spots can be constructed effectively by creating nanogaps on or within metal nanostructures[19,22–25]. Although Raman signal amplifcation has been studied on single nanoparticles (e.g., spherical, polyhedral, and nanostar shapes) and aggregates (e.g., dimers and trimers) with varying sizes and surface morphologies[26,27], most structures suffer from poor controllability in creating uniform hot spots and generating stable SERS signals. Recently, we reported a strategy in designing and synthesizing gap-enhanced Raman tags with a smooth external shell (S-GERTs for brevity), which are composed of metallic core−shell nanoparticles (NPs) with an internal nanogap and embedded aromatic dithiol (e.g., 1,4-benzenedithiol, 1,4-BDT) Raman reporters[13,28–30]. Compared to the nanogaps randomly formed in NP aggregates, S-GERTs provide strong, uniform and stable SERS signals by embedding Raman molecules in the interior nanogap hot spots[11,12]. More efforts have been recently spent to understand and further improve the performance of these NPs[31–39]. However, the overall Raman signal of these S-GERTs is still not strong enough to reach high-speed imaging. For example, the Raman imaging using S-GERTs still takes more than 40 min over a scanning area of 1 cm × 1 cm[40], which is not conducive to real-time rapid imaging during surgery. Therefore, we need to engineer new SERS tags for more versatile and high-speed imaging purpose.

In this work, we report a version of GERTs with petal-like shell structures (P-GERTs for brevity) for high-speed Raman imaging. By utilization of strong electromagnetic hot spots from interior sub-nanometer gaps and external petal-like shell structures, a larger surface area for molecular immobilization, and a larger Raman cross section of reporter molecules, these sub-100 nm sized P-GERTs show two orders of magnitude stronger Raman signals than S-GERTs. P-GERTs can reach a Raman enhancement factor (EF) beyond $5 \times 10^9$ and a detection sensitivity down to a single-NP level. With the fact that the amount of Raman reporters (4-nitrobenzenethiol, 4-NBT) on gold (Au) cores is a key to control shell morphology and SERS performance of P-GERTs, we have suggested a potential growth mechanism of forming petal-like shell. Due to the ultra-strong Raman signal of P-GERTs, ultrafast imaging system and data processing method, high-speed and high-resolution cell imaging (2500 pixels) can be

obtained within 6 s, and high-contrast wide-area ($3.2 \times 2.8$ cm$^2$) in vivo sentinel lymph node (SLN) imaging can be obtained within 52 s. Additionally, a variety of P-GERTs can be obtained to realize multiplexed cell imaging by facilely changing Raman reporters in the external nanogaps. These P-GERTs offer a potential solution to overcome the current bottleneck in the field of SERS-based bioimaging.

## Results

**P-GERTs vs. S-GERTs**. S-GERTs and P-GERTs are both synthesized in a similar wet chemistry process but exhibit quite distinct particle morphology, far- and near-field plasmonic properties, and SERS performance due to the decoration with different interior Raman reporters. It is shown from the schematic diagrams that core-shell structured S-GERTs exhibit a continuous interior nanogap and a smooth external shell surface, where both Raman reporters 1,4-BDT can be adsorbed (Fig. 1c). In contrast, P-GERTs exhibit a continuous internal nanogap but a petal-like shell structure with a rough surface, both decorated with 4-NBT reporters (Fig. 1a). 4-NBT has a similar molecular structure with 1,4-BDT except the nitro group replacing the second thiol group in 1,4-BDT. In a typical experiment of preparing P-GERTs, 22 nm uniform-sized Au cores were modified with 4-NBT molecules via Au−S bonds and were centrifuged to remove the excess amount of molecules, and then the obtained 4-NBT modified Au cores were utilized as seeds for further growth of the Au shell. TEM images confirm that P-GERTs consist of a Au core-rough shell structure spaced by a uniform interior gap with a typical size of ~0.7 nm (indicated by a red arrow in Fig. 1b), which is determined by the thickness of the monolayer of embedded 4-NBT molecules[15,29]. Unlike the S-GERTs with a complete and smooth external Au shell (Fig. 1d)[2], a plenty of nanogaps (indicated by yellow arrows in Fig. 1e) with a size of 1−3 nm are formed in the rough petal-like structures of the Au shell of P-GERTs, denoted as external nanogaps, which are native electromagnetic hot spots for the Raman enhancement. Thus 4-NBT molecules are further decorated on the external nanogaps via a self-assembly process. The overall diameter of P-GERTs is 66 ± 4 nm with a monodispersed particle size and morphology (Fig. 1e).

Aqueous P-GERTs show a dark blue color with a single resonance peak at 600 nm with a relatively broad linewidth, while aqueous S-GERTs show a typical ruby color with a single pronounced resonance peak at 540 nm with a much narrower linewidth (Fig. 1f). Compared with S-GERTs, the resonance peak of P-GERTs is redshifted and becomes broadened most likely due to the strong plasmonic coupling of petal-like Au structures of the rough shell. This also demonstrates a great number of electromagnetic hot spots formed on the external shell of P-GERTs. We then turn to compare the SERS properties of P-GERTs and S-GERTs. It can be seen that the SERS spectrum of P-GERTs when excited by 638 nm laser exhibits characteristic Raman bands dominated by the strong mode of $v$ (NO$_2$) at 1340 cm$^{-1}$ and four relatively weak modes by $\pi$ (CH) + $\pi$ (CS) + $\pi$ (CC) at 723 cm$^{-1}$, $\pi$ (CH) at 854 cm$^{-1}$, $v$ (CS) at 1083 cm$^{-1}$, and $v$ (CC) at 1575 cm$^{-1}$ (Fig. 1g). While S-GERTs exhibit two relatively strong Raman bands at 1055 and 1555 cm$^{-1}$ and one weak band at 1178 cm$^{-1}$ [28,41,42]. More importantly, we found that the total Raman intensity of P-GERTs is more than two orders of magnitude higher than that of S-GERTs under the same experimental condition, estimated by comparing the Raman band at 1340 cm$^{-1}$ for 4-NBT and at 1055 cm$^{-1}$ for 1,4-BDT. It is also found that the Raman intensity of P-GERTs is roughly one order larger than that of the DNA-bridged SERS tags previously reported[17], after

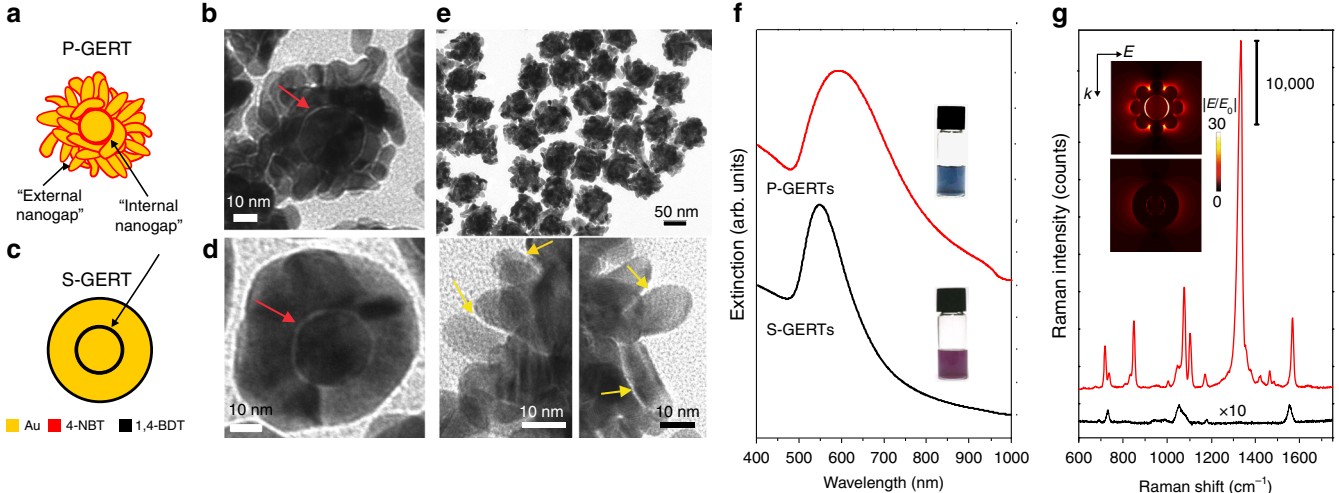

**Fig. 1** The comparison of P-GERTs and S-GERTs. Schematic diagrams and representative TEM images of GERTs with a petal-like shell (P-GERTs, **a**, **b**), and GERTs with a smooth-surface shell (S-GERTs, **c**, **d**). Red arrows in panel **b**, **d** indicate the internal nanogaps of GERTs. The scale bars are 10 nm. **e** Representative low-resolution TEM image (top, the scale bar is 50 nm) and high-resolution TEM images (bottom, the scale bar is 10 nm) of P-GERTs. Yellow arrows in panel **e** indicate the external nanogaps between petal-like structures on the shell of P-GERTs. **f** Extinction spectra of P-GERTs (red) and S-GERTs (black). The insets in panel **f** show the corresponding photos of GERTs solution. **g** SERS spectra of P-GERTs (red) and S-GERTs (black). The insets in panel **g** show the finite-difference time-domain (FDTD) calculated electric field enhancement distribution of a single P-GERT (top) and S-GERT (bottom) excited by 638 nm. Raman measurements were performed using 638 nm laser, 2 s acquisition time, and ×10 objective lens

normalizing all experimental parameters (see more details in Supplementary Note 4). Different SERS performances can be explained by the finite-difference time-domain (FDTD) calculated electro-magnetic field distributions (insets in Fig. 1g). It can be found that the internal and external nanogaps of the P-GERT both produce quite strong electromagnetic field enhancement when excited by 638 nm laser, which can greatly enhance the Raman signal. But the S-GERT only shows the hot spots in the internal nanogap, and the electromagnetic field enhancement on the external shell is much smaller. We presume that the following three factors have contributed to the ultrahigh Raman intensity of P-GERTs. First, the internal nanogap still exists in P-GERTs, and the plasmon coupling between the Au core and shell produces strong electromagnetic enhancement in internal nanogap region. The petal-like shell structure of P-GERTs is more conducive to light penetration than the complete shell of S-GERTs, resulting in more efficient light excitation for the internal nanogaps. Second, the larger surface area and multiple external nanogaps in P-GERTs offer a larger number of immobilization site and hot spots for reporter molecules, therefore achieving a strong Raman enhancement[43]. Third, 4-NBT molecules have a larger Raman cross section than 1,4-BDT due to the intrinsic chemical structure[44].

To investigate the enhancement contribution of the external hot spots to the total Raman enhancement of P-GERTs, we compared the Raman spectra of P-GERTs before and after the decoration of 4-NBT molecules in the external nanogaps when excited by three different laser wavelengths (532, 638, and 785 nm) with a particle concentration of 0.2 nM. The 2 nm redshift of the resonance peak of P-GERTs suggests that the 4-NBT molecules adsorbed successfully in the external nanogaps (Supplementary Fig. 1). P-GERTs show remarkable SERS signals for all three excitation wavelengths, and they in general show a trend with the strongest signal at 638 nm, relatively weak at 785 nm, and the weakest at 532 nm no matter before or after the external decoration of 4-NBT molecules (Supplementary Fig. 2a). Moreover, they all show the Raman intensity improvement after the external decoration of 4-NBT molecules at three wavelengths,

for example, 19%, 47%, and 21% for 532, 638, and 785 nm, respectively. In contrast, S-GERTs exhibit much weaker Raman signals under the same three excitation wavelengths, and the functionalization of Raman reporters (1,4-BDT) on the external surface only provides 16%, 8.2%, and 10% intensity improvement for 532, 638, and 785 nm excitation (Supplementary Fig. 2b). Compared with S-GERTs, the SERS intensities of P-GERTs were around 160-, 128-, and 86-fold higher with the contribution of external hot spots for 532, 638, and 785 nm laser excitation, respectively (Supplementary Fig. 2c). The SERS enhancement factors (EFs) of P-GERTs are estimated to be $8.7 \times 10^8$, $5.4 \times 10^9$, and $3.7 \times 10^9$ for 532, 638, and 785 nm, respectively (see detailed calculations in Methods). These results confirm that the external hot spots play an important role for the Raman enhancement of P-GERTs, and also demonstrate the potential of P-GERTs as multiple-wavelength active SERS probes. The FDTD calculations additionally disclose that the electromagnetic field both inside the internal and external nanogaps of P-GERTs exhibit the strongest enhancement with 638 nm excitation compared to 532 and 785 nm excitation (Supplementary Fig. 2d), which is consistent with the experimental results.

**Morphology and optical tuning of P-GERTs.** In our previous work of S-GERTs[45], we have found that a longer incubation time of 1,4-BDT reporters with Au cores can form a multilayered molecular structure, which affects the morphology (e.g., the thickness) of the internal nanogap and the consequent optical properties of GERTs, but their external Au shell remains unaffected with a spherical shape and a smooth surface. Herein, 4-NBT molecules were also self-assembled on Au cores via Au−S covalent bonding but cannot form a multilayered molecular structure due to the lack of the second thiol group even when a large excess of 4-NBT was added[45,46]. After a super-long incubation time of 960 min for Au cores and 4-NBT, only a 2.6-nm redshift of the plasmon resonance peak was observed (Supplementary Fig. 3), which further confirms that 4-NBT molecules can only form a monolayer on Au cores[46]. However, when we varied the incubation time of the Au cores in 4-NBT (10 mM)

solution from 0.5 to 960 min, it exhibited great impact on the surface morphologies and optical properties of P-GERTs. When the incubation time is 0.5 min, the Au shell is a complete piece with a smooth external surface that is very similar to S-GERTs, but only an incomplete internal nanogap can be observed (Fig. 2a). When the incubation time increases to 2 min, the Au shell is still intact but with a relatively complete interior nanogap (Fig. 2a). Further increasing of incubation time from 5 to 60 min, the Au shell starts to become petal-like still with a continuous interior nanogap and multiple external nanogaps (Fig. 2a). The number of petal-like structures increases with the extension of incubation time. After 960 min incubation, the Au shell is composed of many small petal-like structures, and the identification of interior nanogaps become more difficult (Fig. 2a and Supplementary Fig. 4). TEM measurements of a large number of P-GERTs with various incubation time show that they all have a monodispersed particle size roughly in the range of 60−70 nm (Supplementary Fig. 5).

To understand how the number of 4-NBT molecules on Au cores influences the growth of Au shells, we investigated their growth process using time-lapse TEM analysis. Snapshots of intermediate products were obtained by terminating the Au shell growth process at different time points (from 1 to 12 min) after 4-NBT modified Au cores were introduced into the reaction. When the incubation time is 0.5 and 2 min, 4-NBT exhibits an adsorption density of around 2.7 and 3.3 molecules/$nm^2$ on Au cores, according to its molecular footprint of ~0.2 $nm^2$ and the plasmon resonance shifts in Supplementary Fig. 3[47], and they primarily lie down on the Au core surface and form a leaky monolayer (Fig. 2b)[15]. The Au shell growth starts with a relatively large nucleation on the core, and gradually turns outward to form the shell and wrap around the Au core[45]. Since the 4-NBT molecules are in lying-down orientation and the nitro group has a small contact area with the Au shell, the Au shell can easily form a complete piece which is similar to S-GERTs. Further increasing of incubation time to 5–960 min allows the improvement of the adsorption density to 4–5 molecules/$nm^2$, and 4-NBT molecules form a more densely packed monolayer on the cores in a more vertical state (Fig. 2c). Due to the poor affinity between Au and the nitro group, the Au shell can only grow from a number of small nucleation sites on cores, and gradually forms petal-like structures, which is more conducive to light penetration[31]. In addition, the internal nanogaps also become more discontinuous. Time-lapse TEM examination offers us a preliminary understanding of the petal-like shell growth mechanism, but a more advanced characterization such as high-resolution in situ TEM may further disclose more insights.

Next, we optimize GERTs for high-speed Raman imaging in terms of far-field extinction spectra and near-field SERS properties for the P-GERTs with different morphologies after further decoration of 4-NBT molecules into the external nanogaps. Figure 2d indicates that the contribution of the Raman enhancement from the internal and external near-field hot spots (shaded area) becomes significant when the incubation time is longer than 2 min (namely, after forming petal-like shell structures) for both 633 and 785 nm lasers excitation (see more details in Supplementary Fig. 6). This is obviously due to the formation of near-field hot spots in the internal and external nanogaps. However, S-GERTs with their relatively smooth shells show little signal enhancement effect after adsorbtion of reporter molecules on their shells (Supplementary Fig. 2b, c). This optimization process of the morphology and SERS properties illustrates that P-GERTs with 10 min incubation time show the optimal Raman intensity, and we proceed in the later experiments using this sample unless otherwise stated. By considering a larger surface area of external shell and a larger molecular Raman cross

section, we have also found that the Raman enhancement factor from the external hot spots is roughly one order of magnitude smaller than that from the internal hot spots in P-GERTs although the overall Raman signal is dramatically enlarged. This is probably explained by the difficulty of molecular diffusion into the external nanogaps during the self-assembly process.

**Single-NP detection of P-GERTs**. Single-NP SERS detection raises new potential for multiplexed molecular diagnosis and in vivo Raman spectroscopy and imaging[48–50]. Therefore, we further challenge the sensitivity and the photostability of P-GERTs down to a single-NP level. Before the measurement, we first coated the P-GERTs with a mesoporous silica layer (MS P-GERTs for brevity) to prevent the formation of inter-particle electromagnetic hot spots (Fig. 3a). The MS layer showed a homogeneous thickness of 13 ± 2.4 nm, which is thick enough to screen the inter-particle plasmon coupling[51,52]. Fig. 3b shows the representative SERS spectra obtained from aqueous MS P-GERTs with different concentrations of 10 pM, 1 pM, 100 fM, 10 fM, and 1 fM excited by 638 nm laser, indicating clear 4-NBT Raman profiles with a good signal-to-noise ratio even at the lowest concentration of 1 fM (Supplementary Fig. 7). We also notice that we could constantly detect Raman signals from the MS P-GERTs for relatively high concentrations, such as 10 pM, 1 pM, and 100 fM, but only signals in about a quarter of measurements for the concentration of 10 fM, and signals in very few cases for the concentration of 1 fM. Then we performed Raman measurements in a time sequence with P-GERTs concentration of 100, 10, and 1 fM in a more accurate way. The probed volume was estimated to be approximately 32 pL according to the method described previously[53,54], corresponding to an average of 2, 0.2, and 0.02 NP in the probed volume when the concentration is 100, 10, and 1 fM, respectively. With such few particles, the average residence time of a particle in the probed volume can be roughly estimated to be 10–20 s due to the Brownian motion into and out of the probed volume[53], so we set the measurement time as 10 s. Signals measured from a sample without GERTs were used to establish the background threshold of 117 counts, which is three times the standard variation in the mean background signal (indicated by blue dotted lines in Fig. 3d and Supplementary Fig. 8).[31] Fig. 3c–e display SERS spectra, band intensity and statistical analysis (1340 $cm^{-1}$) of 120 Raman measurements in a time sequence with an average of 2, 0.2, and 0.02 NP per probed volume. For the case of 2 particles per volume, all 120 measurements exhibited pronounced Raman spectra (Fig. 3c top) with the band signal (1340 $cm^{-1}$) beyond the threshold (Fig. 3d top), and the corresponding statistical analysis shows a Gaussian statistical distribution of the frequency of the appearance of the Raman signal (Fig. 3e top). When the particle concentration is decreased to an average of 0.2 particles per volume, only ~40% of measurements meet the threshold criteria (middle in Fig. 3c, d), and the SERS signal distribution exhibits four relative maxima that reflect the probability of finding 0, 1, 2 or 3 NPs in the proved volume, respectively. The distribution can be reasonably fit by the superposition of four Gaussian curves, which is roughly consisting with a Poisson distribution (Fig. 3e middle). The results obtained with an average of 0.02 NP per volume further illustrate that only ~3% measurements show acceptable Raman signals, with three fitted Gaussian curves on the intensity distribution that reflect the probability of finding 0, 1, or 2 NPs in the proved volume. This is roughly consistent with a Poisson distribution as well. When the average number of particles in probed volume is 1 or less, the change in the statistical distribution of SERS signals from Gaussian to Poisson distribution provides evidence for the single-NP detection by SERS[53]. It can be also noticed that the

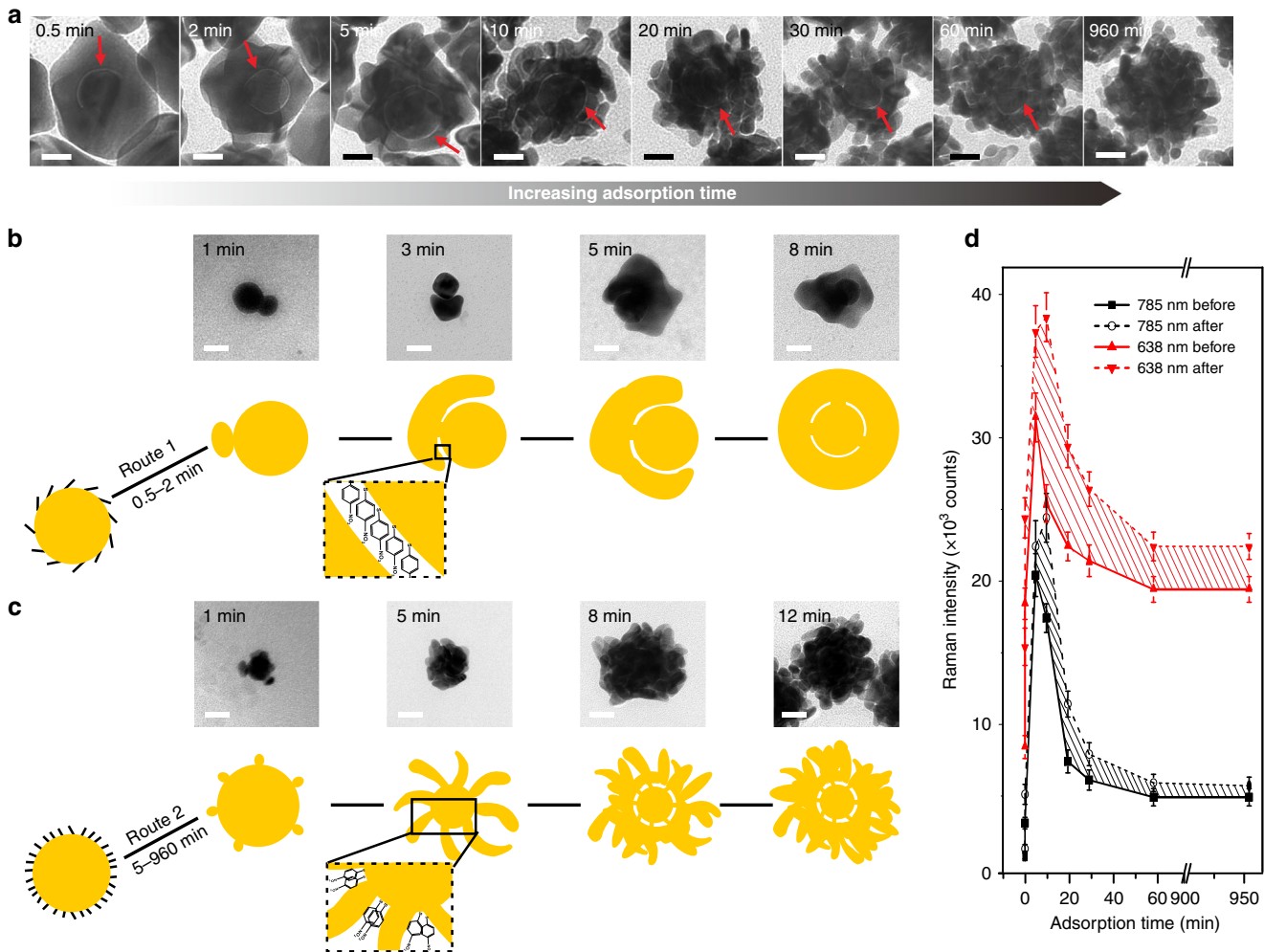

**Fig. 2** Highly tunable morphology of P-GERTs. **a** Representative TEM images of P-GERTs as a function of incubation time (0.5, 2, 5, 10, 20, 30, 60, and 960 min, respectively) of Au core and Raman reporter 4-NBT. Red arrows indicate the internal nanogaps of P-GERTs. Presumed growth process of the Au shell on **b** low-density and **c** high-density 4-NBT decorated Au cores, with corresponding time-lapse TEM examinations. All scale bars are 20 nm. **d** Comparison of mean SERS intensity before and after 4-NBT molecules decoration on the external surface of P-GERTs with various morphologies excited by 785 nm (black) and 638 nm (red) laser. Error bars are standard deviations from three independent measurements. Source data are provided as a Source Data file

Raman measurements of ultra-low concentration of aqueous P-GERTs excited by 785 nm show quite similar behaviors to those by 638 nm, only with a slightly lower (around 70%) Raman intensity (Supplementary Fig. 9). Thus, we conclude the detection sensitivity of P-GERTs down to a single-NP level in liquid state. We have additionally performed the stability analysis of aqueous MS P-GERTs. The extinction and Raman spectra in aqueous solution of different pH values (from 5 to 10) and in 10% fetal bovine serum (FBS) and saline for 72 h show stable intensities and unchanged spectral profiles (Supplementary Fig. 10). These results indicate excellent materials stability of aqueous P-GERTs suitable for biomedical applications in different environments.

We further examined the single-NP Raman signal of MS P-GERTs in solid state by atomic force microscope (AFM)-correlated nano-Raman measurement to quantify their signal uniformity and photostability[19]. It should be noted that we used the dielectric Si tips in the nano-Raman measurement (Fig. 4a), which is obviously distinct from tip-enhanced Raman scattering (TERS) measurement. The MS P-GERTs were coated with a MS layer with a thickness of 13 nm so that the tip end cannot directly contact with 4-NBT molecules. NP solution was first spin-coated on a silicon wafer and dried, and then the laser focal spot was exactly matched with the center of the AFM Si tip for symmetrical scattering on the Si tip end. The Raman measurement could be performed in a contact or non-contact mode. A typical non-contact mode AFM image gives a confirmation of the formation of isolated single NPs on the substrate with a particle height of ~96 nm (Fig. 4b). Figure 4c displays the correlated SERS spectra from the corresponding three particles (particle 1−3) in Fig. 4b measured in contact and non-contact modes. These three particles all show very strong and uniform Raman signals of 4-NBT molecules (~450 counts) in the contact mode, which is about 10% stronger than that in the non-contact mode. This may be because the tip can assist the focus of light better onto the particles in the contact mode. More than 80% of 34 single-NP measurements produced Raman signals between 350 and 450 counts, demonstrating a quite good uniformity at the single-NP level (Fig. 4d). Although the petal-like structures of P-GERTs are random and poorly controlled, each P-GERT has a large number of petal-like structures on the surface, which results in an average Raman signal of each NP in a decent uniformity. More importantly, single-NP photostability examinations of P-GERTs

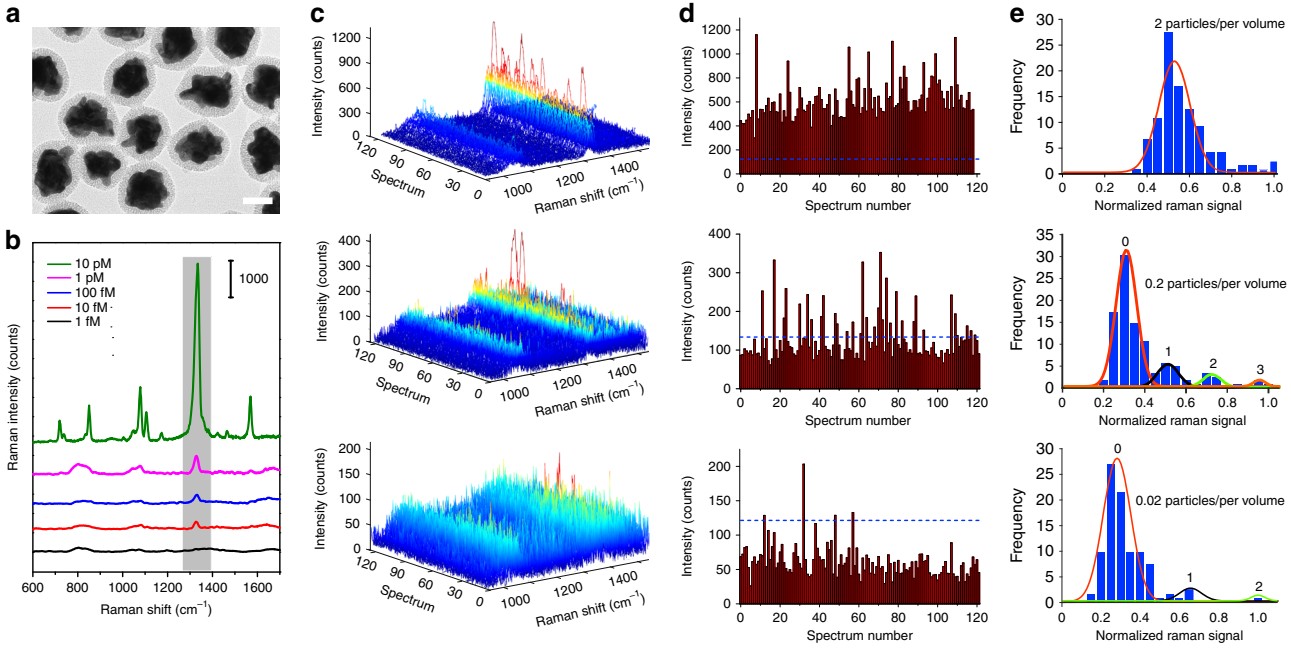

**Fig. 3** Single-NP detection of aqueous P-GERT solution. **a** TEM image of mesoporous silica-coated (MS) P-GERTs. Scale bar is 20 nm. **b** Concentration-dependent Raman measurements of aqueous MS P-GERTs (638 nm laser, 20 mW, 10 s acquisition time, and ×60 objective lens). **c** SERS spectra plot, **d** the intensity of Raman band at 1340 cm$^{-1}$, and **e** the corresponding statistical analysis of 120 measurements from 100 (top), 10 (middle), and 1 (bottom) fM P-GERT solution. 100, 10, and 1 fM GERT solutions correspond to 2, 0.2, and 0.02 particle per probing volume, respectively. Blue dotted lines in panel **d** represent the background Raman signal (1340 cm$^{-1}$) without P-GERTs. All solid curves in panel **e** are Gaussian fitting to the data. The fitted peaks show the Gaussian (top) or Poisson (middle and bottom) distribution of the data. For Poisson distribution, the red, black, green and orange peaks reflect the probability to find 0, 1, 2, or 3 particles per volume

indicate that their SERS signals remain quite stable after continuous laser irradiation (110 μW) for 1800 s (Fig. 4e). The inset in Fig. 4e shows all time-resolved spectra of a single NP during continuous irradiation without significant fluctuation or appearance of new Raman bands. Their super-stable SERS properties are further proved by the almost negligible change of the three representative SERS spectra measured before, during, and after irradiation (Fig. 4f). All these results imply that P-GERTs exhibit a large Raman enhancement down to a single-NP detection sensitivity, decent particle-to-particle SERS uniformity, and ultrahigh photostability.

**High-speed and high-contrast Raman cell and tissue imaging.** It typically takes minutes to hours to acquire a Raman cell image with SERS tags[55,56]. We have recently demonstrated that S-GERTs-based cell imaging with a resolution of 50 × 50 pixels could be obtained within 35 s with a much higher laser power of 3.7 mW[30]. Such large Raman enhancement of P-GERTs down to a single-particle level may further allows us to improve the imaging speed with a further lower laser power, which is crucial to minimize the photodamage to biological samples. We performed a high-speed Raman cell imaging in a step-by-step mapping mode, where the laser is scanned across the cell in X and Y directions by the rotation of galvo mirrors instead of moving the stage mechanically (see Fig. 5a). As a demonstration, Fig. 5b–d show the bright-field image, Raman image and overlay of a fixed H1299 cell passively stained with the P-GERTs. The bright-field image and Raman image of the cell coincided well. The high-resolution single-cell Raman image (50 × 50 pixels for 55 × 66 μm$^2$ field of view) could be acquired within 6 s with a laser power 370 μW and an ultrashort acquisition time of 0.7 ms per pixel (see more details in Methods). 0.7 ms is so far the shortest acquisition time of commercial electron-multiplying charge-

coupled device (EMCCD) to the best of our knowledge. It allows to largely improve the imaging speed one to two orders of magnitude faster compared to the previous work performed with a similar image resolution and laser power[4]. Raman imaging of multiple cells in a larger area (61 × 41 pixels for 380 × 240 μm$^2$ field of view) can also be accomplished within 7 s (Supplementary Fig. 11). Such high imaging speed can be attributed to the significant improvements in SERS tag signal, imaging system, and data processing method. Firstly, P-GERTs with multiple electro-magnetic hot spots can provide sufficient Raman signals, which greatly shorten the signal acquisition time. Secondly, the step-by-step imaging mode realizes rapid movement of the laser spot by the rotation of galvo mirrors, which greatly reduces the scanning time. Thirdly, data processing line-by-line in parallel in SWIFT mode also helps time saving[57–59]. More importantly, three Raman spectra randomly selected from different parts (point 1–3 in Fig. 5c) of the cell clearly exhibit characteristic bands of 4-NBT molecules with a rather good signal-to-noise ratio within such an ultrashort acquisition time of 0.7 ms per pixel (Fig. 5e). These result in the possibility to detect a single cell labelled with less than 100 P-GERTs based on the aforementioned single-NP AFM nano-Raman measurements (refer to Supplementary Fig. 12 for more details). In addition, the background Raman and auto-fluorescence signals from the cell and the culture dish are negligible within such a short exposure time, consequently resulting in an ultrahigh-contrast Raman image (Fig. 5c). It can be also seen from Fig. 5b that the addition of nanoprobes did not generate a significant effect on the morphology of cell, indicating P-GERTs may have good biocompatibility. A CCK-8 assay test further proved that the P-GERTs are nearly non-cytotoxic to cells within a concentration range of 0−1 nM (Supplementary Fig. 13), implying great potential for in vitro and in vivo bioimaging. We additionally demonstrated that P-GERTs exhibited excellent

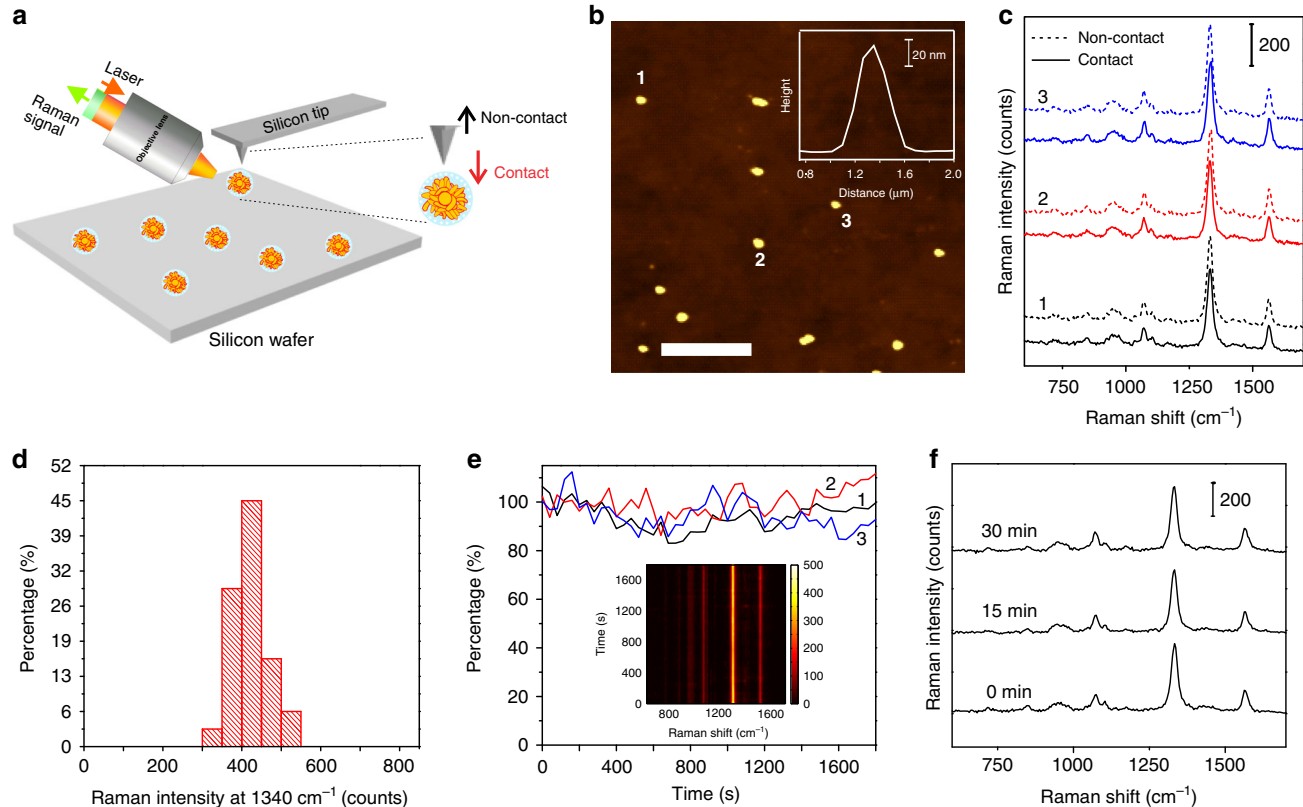

**Fig. 4** AFM-correlated single-NP nano-Raman analysis of the P-GERTs. **a** Schematic illustration of AFM-correlated nano-Raman spectroscopy. **b** A representative AFM image of MS P-GERTs on silicon wafer. The scale bar is 5 μm. The inset shows a typical cross-section profile of a single MS P-GERT. **c** SERS spectra of three single MS GERTs (particle 1, 2, and 3) indicated in panel **b** measured in contact and non-contact modes. **d** Distribution diagram of the SERS intensity (1340 cm⁻¹) of thirty-four single MS GERTs from different areas on silicon wafer. **e** Photostability measurements of three single MS GERTs under continuous laser irradiation for 30 min, monitored by the integrated band area at 1340 cm⁻¹. The inset shows a typical time-resolved SERS spectral map of a single GERT during continuous irradiation for 30 min. **f** Three representative SERS spectra at selected irradiation times under irradiation from the same particle. All measurements were performed with a 633 nm laser, 110 μW power, 10 s acquisition time, and ×100 objective lens

long-term photostability during high-speed 30-min continuous cell imaging process (Supplementary Fig. 14) and the great potential for high-speed live-cell imaging (Supplementary Fig. 15). Therefore, P-GERTs provide various advantages for Raman cell imaging such as decreased incident laser power, high spatial and temporal resolution, high-contrast image quality, and long-term photostability.

The sentinel lymph nodes (SLNs) are the theoretical first site of cancer metastasis, and the accurate locating of SLNs is fundamental for performing sentinel lymph node biopsy (SLNB) to determine lymphatic spread of cancer cells. To demonstrate the potential of P-GERTs for in vivo imaging, we performed Raman imaging of the SLN in mice. The mice ($n = 3$) were injected subcutaneously with P-GERTs in saline (25 μL, 0.5 mg/mL) in the paw of the left hind limb and the popliteal lymph node was modeled as the SLN. After 24 h, the mice were anesthetized and high-speed Raman imaging of SLNs were performed in a Macro mapping mode (see Fig. 5f), where the laser beam is raster-scanned to record an average Raman spectrum across an area sub-millimeter by sub-millimeter and the sample is moved by the motorized XY stage with a step to cover the whole surface. Compared to the traditional motorized stage movement with a normal laser spot (typically several micrometers in diameter), we can minimize the time spent on stage response and movement as much as possible without losing effective Raman information in the Macro mapping mode. Raman images in Fig. 5g, h show strong SERS signals of P-GERTs from the SLN and the injection

site, while negligible signal from the surrounding tissues, indicating that P-GERTs can be specifically enriched in the SLNs by lymphatic circulation to trace SLNs. The wide-area in vivo Raman image ($3.2 \times 2.8$ cm²) was obtained within 52 s ($20 \times 20$ pixels) with a laser power of 370 μW and 0.7 ms acquisition time per pixel. During the SLN imaging process, it is worth noting that each pixel contains all Raman signals over an area of $600 \times 600$ μm², which usually requires hundreds of seconds in normal stage movement mode. The Macro mapping mode enables us to collect sample information as much as possible with a short imaging time and few pixels.

The Raman spectra from SLN (point 2), adjacent tissues (point 1) and the surrounding environment (point 3) further confirms the specificity of P-GERTs for lymph node tracing (inset in Fig. 5h). Because the signal acquisition time is only 0.7 ms, the signal generated by tissue is not obvious and similar to the blank background. Figure 5i shows cross sections of the Raman signal intensity (1340 cm⁻¹) and the signal-to-background ratio (SBR) from the Raman image, marked by a white dashed line in panel h of Fig. 5h. The Raman signal of the surrounding tissues is considered as the background. Although the acquisition time per pixel is only 0.7 ms, the Raman signal acquired in the SLN region reaches a SBR of around 6.5. Unlike fluorescent nanoprobes, P-GERTs have a unique fingerprint Raman signal from 4-NBT molecules, which is not available from biological tissues. Therefore the raw Raman spectrum (for example, point 2 in the inset of panel h) can be further processed to remove the background

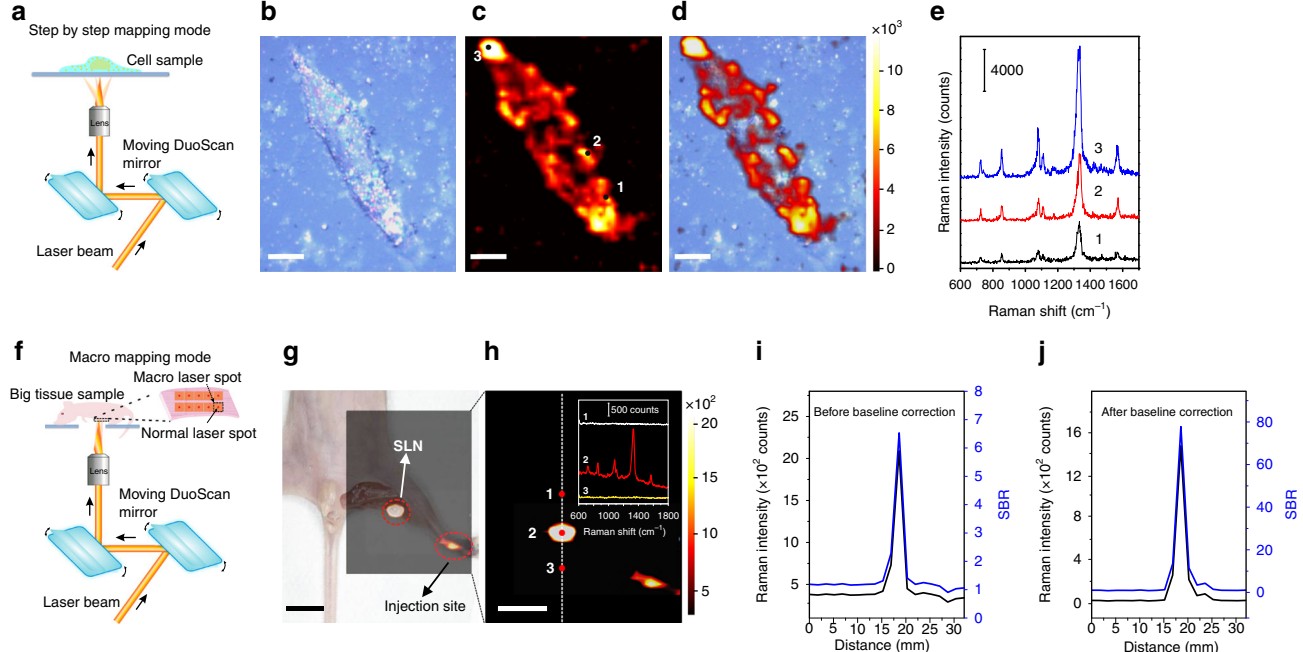

**Fig. 5** High-speed and high-contrast wide-area Raman imaging. **a** Schematic illustration of the stepping scan mode for high-speed cell Raman imaging. **b** Bright-field image, **c** Raman image, and **d** the overlay image of a single H1299 cell. The Raman image is obtained within 6 s (50 × 50 pixels) with a laser power of 370 μW and a 0.7 ms acquisition time per pixel. Scale bars in panel b–d are 10 μm. **e** Three representative SERS spectra obtained from different points in panel **c**. **f** Schematic illustration of the Macro scan mode for wide-area tissue imaging. (**g**, **h**) High-contrast and wide-area in vivo Raman image (3.2 × 2.8 cm²) of the hind-limb popliteal lymph node after injection of P-GERTs. The Raman image was obtained within 52 s (20 × 20 pixels) with a laser power of 370 μW and acquisition time of 0.7 ms per pixel. Scale bars in panel **g** and **h** are 1 cm. The inset in **h** shows Raman spectra at the SLN in situ (point 2), adjacent tissue (point 1) and surrounding environment (point 3). All Raman images are plotted using the Raman band at 1340 cm⁻¹. The intensity and signal-to-background ratio (SBR) of the Raman signal (1340 cm⁻¹) cross-section from top to bottom marked by the white dashed line in **h** (**i**) before and **j** after the baseline correction of Raman spectra

signals such as autofluorescence from adjacent tissues. The SBR of the Raman in vivo imaging can be greatly improved by an order of magnitude to around 80 (see Fig. 5j) after the baseline correction of the raw Raman spectra, realizing extremely high-contrast in vivo imaging. The enrichment of P-GERTs in the SLN was also confirmed by imaging a dissected lymph node (Supplementary Fig. 16). With the advantageous features including strong Raman enhancement, unique fingerprint Raman signal, high photostability, and good biocompatibility, P-GERTs hold the great potential to be used for high-speed and high-contrast intraoperative wide-area Raman imaging.

**Multiplexed Raman cell imaging**. By replacing Raman reporters immobilized in the external nanogaps of P-GERTs, we can further demonstrate their multiplexing imaging capability. We introduced four kinds of molecules including 3,4-dichlorobenzenethiol (3,4-DBT), 4-mercaptobenzonitrile (4-MBN), 3-fluorothiophenol (3-FTP), and 2-naphthalenethiol (2-NT) into the external nanogaps of the petal-like shell of P-GERTs (Fig. 6a). These four kinds of molecules are chosen because they all have a thiol group, forming a covalent bonding to the Au surface of P-GERTs, and each molecule has at least one characteristic Raman band that can be better distinguished from 4-NBT molecule. We obtained Raman spectra of P-GERTs with these four kinds of reporters with different characteristic bands (Fig. 6b), indicating the unique characteristic band of 2230 cm⁻¹ for 4-MBN (purple), 1001 cm⁻¹ for 3-FTP (green), 568 cm⁻¹ for 3,4-DBT (red), and 1379 cm⁻¹ for 2-NT (blue). The cells were incubated with the mixture of four kinds of GERTs (1:1:1:1 for molar concentration) for 6 h, and then multiplexed Raman

imaging was performed. A multiplexed Raman image of a single cell (50 × 50 pixels for 24 × 66 μm² field of view) could be acquired within 43 s with a laser power of 370 μW and 10 ms acquisition time per pixel. As shown in Fig. 6c, d, the Raman cell image reconstructed from the characteristic band of 1340 cm⁻¹ of 4-NBT, the common reporters embedded in the internal nanogaps of all four kinds of GERTs, were well consistent with the bright-field image. The Raman images reconstructed using the unique characteristic bands of four external reporters clearly show the slightly different distribution of each kind of GERTs in the cell (Fig. 6e). The multiplexed encoding ability of P-GERTs makes it potential for imaging nanoprobes targeting to different sites of cells or different cells.

## Discussion

In summary, we have developed P-GERTs with 4-NBT Raman reporters embedded in electromagnetic hot spots from the interior nanogaps in the core-shell junction and external nanogaps in the petal-like shell structure for high-speed, high-contrast cell and tissue bioimaging. P-GERTs exhibit around two orders of magnitude stronger SERS signals than S-GERTs at all three laser excitation wavelengths of 532, 638, and 785 nm, reaching a maximum Raman EF beyond 5 × 10⁹. The change in the statistical distribution of SERS signals from Gaussian to Poisson distribution from continuously decreased concentration of aqueous P-GERTs demonstrates a detection sensitivity of the single-NP level. The nano-Raman measurements confirm the single-NP sensitivity in solid state and additionally show decent NP uniformity and photostability, evaluated at a single-NP level. The high SERS signal improvements of P-GERTs are attributed to the larger

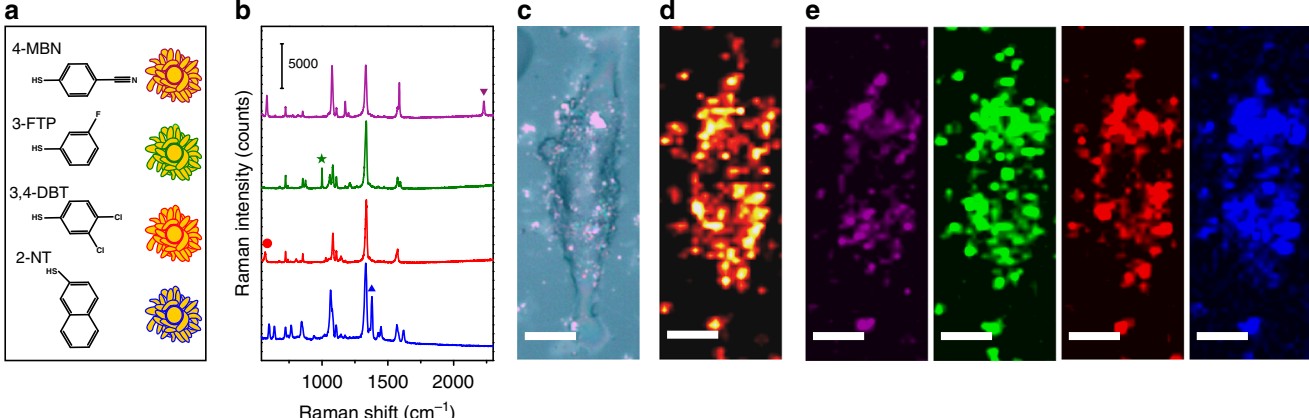

**Fig. 6** Multiplexed Raman imaging of a H1299 cell. **a** Schematics and **b** SERS spectra of P-GERTs with different Raman reporters of 4-MBN (purple), 3-FTP (green), 3,4-DBT (red), and 2-NT (blue) immobilized in the external nanogaps. **c** Bright-field image and (**d**, **e**) multiplexed SERS cell imaging of a single H1299 cell. Raman image in panel **d** was plotted from 1340 cm⁻¹ of embedded 4-NBT reporter and images in **e** were plotted from the Raman signals from external Raman reporters 4-MBN (purple), 3-FTP (green), 3,4-DBT (red), and 2-NT (blue), respectively. All scale bars are 10 μm

surface area for molecular decoration, large Raman scattering cross-section of 4-NBT molecules and the difference in shell morphology, where S-GERTs are in a complete piece with a smooth surface while P-GERTs have a petal-like structure with multiple nanogaps. By adjusting the amount of 4-NBT adsorbed on Au cores, we can regulate the morphology and SERS performance of P-GERTs. By changing the type of external Raman reporters, a variety of probes can be obtained to achieve multiplexed detection and imaging. Due to the ultra-strong Raman signal of P-GERTs, ultra-fast laser scanning technique and data processing method, we have achieved high-speed high-resolution single-cell Raman imaging (50 × 50 pixels for 55 × 66 μm² field of view) acquired within 6 s with a laser power 370 μW and acquisition time of 0.7 ms/pixel, and high-contrast wide-area in vivo SLN Raman imaging (3.2 × 2.8 cm²) obtained within 52 s with a laser power of 370 μW, acquisition time of 0.7 ms/pixel, and the SBR of around 80. P-GERTs may open new opportunities for rapid ultrasensitive biosensing and bioimaging platform as superbright and highly stable optical nanolabels.

## Methods

**Materials**. All materials were used as received without any further purification. Chloroauric chloride (HAuCl₄·4H₂O), ascorbic acid, tetraethyl orthosilicate (TEOS), sodium hydroxide (NaOH), and methanol were purchased from Sinopharm Chemical Reagent Co. Ltd (Shanghai, China). Sodium borohydride (NaBH₄, 98%), cetyltrimethylammonium chloride (CTAC, 99%), 3, 4-dichlorobenzenethiol (3,4-DBT, 98%), 2-naphthalenethiol (2-NT, 98%), 3-fluorothiophenol (3-FTP, 98%) and 4-mercaptobenzonitrile (4-MBN, 95%) were obtained from J&K Chemical Ltd (Shanghai, China). 4-Nitrothiophenol (4-NBT, 90%) was purchased from Fluorochem Ltd (Derbyshire, United Kingdom). 1,4-Benzenedithiol (1,4-BDT, 98%) was received from TCI (Tokyo, Japan). The ultra-fine insulin syringes (1 mL) were purchased from Becton, Dickinson and Company (USA). The cell counting kit-8 (CCK-8) was purchased from Dojindo Laboratories (Kumamoto, Japan). DMEM/High glucose medium, penicillin–streptomycin mixture solution, fetal bovine serum (FBS) and 0.25% trypsin-EDTA were purchased from Gibco Life Technologies (USA). Nanopure water (18.2 MΩ) was used for all experiments.

**Synthesis of P-GERTs and S-GERTs**. Au cores were first synthesized using a seed-mediated process described in our previous work[28]. Next, 500 μL of 10 mM 4-NBT ethanol solution was added to Au cores (10 mL) under vigorous sonication. The mixtures were incubated at 25 °C for 0.5, 2, 5, 10, 20, 30, 60, and 960 min, respectively. These 4-NBT modified cores were then resuspended in 0.05 M CTAC solution after washing three times with 0.05 M CTAC solution. After that, various incubation time of 1 mL 4-NBT modified core solution were added into the mixed growth solution of 16 mL CTAC solution (0.05 M), 480 μL of ascorbic acid (0.04 M), and 960 μL of HAuCl₄ (4.86 mM) under vigorous sonication. The color of the solution changed from colorless to pink, purple, and blue, and 4-NBT GERTs (P-GERTs) can be obtained in about 10 min. 1,4-BDT GERTs (S-GERTs) were synthesized according to the procedure described in our previous work[28].

Mesoporous silica coating of 4-NBT GERTs was applied according to Gorelikov and Matsuura's protocol with some modifications[60]. The obtained 4-NBT GERTs (50 mL) were washed three times and then dispersed in 5 mL water. NaOH solution (40 μL, 0.1 M) was added with stirring. Three 50 μL injections of 5% TEOS in methanol were carried under gentle stirring at 30 min intervals, and the reaction was kept for 24 h. The resulting products were collected by centrifugation and washed with ethanol (25 mL) containing NH₄NO₃ (10 mg) for three times.

**Modification of Raman reporters on external nanogaps of P-GERTs**. The obtained 4-NBT GERTs (10 mL) were mixed with 500 μL of 10 mM Raman reporters (4-NBT, 4-MBN, 3-FTP, 3,4-DBT, 2-NT, respectively) under vigorous sonication for 1 h. The samples were then washed for three times by centrifuge at 11000 rpm for 10 min to remove excess reagent, and further dispersed in 5 mL aqueous CTAC solution (0.05 M). Mesoporous silica coating was then applied to the obtained samples.

**Characterization of GERTs**. Transmission electron microscopy (TEM) images were collected on a JEM-2100F TEM (JEOL, Tokyo, Japan) operated at 200 kV. UV-Vis spectra were obtained using a UV1900 UV-Vis spectrophotometer (Aucybest, China). The Raman test was carried out on confocal Raman microscope with 532, 638 and 785 nm laser (Horiba, Xplora INV). The 638 nm laser provided by Horiba was an air cooled CLDS laser diode with a linewidth smaller than 0.03 nm. Nano-Raman measurements were carried out on atomic force microscope (AFM)-correlated Raman system (633 nm, ×100 objective, NA = 0.9). An enzyme-linked immunosorbent assay microplate reader (Synergy 2, Bio-TEK) was used to measure the absorbance. Time-lapse TEM analysis of intermediate products were obtained by terminating the Au shell growth process at different time points (from 1 to 12 min) after 4-NBT modified Au cores were introduced into the reaction. Typically, a holey carbon-coated copper grid was immersed into the reaction mixture and immediately put into liquid nitrogen to stop the reaction. The copper grid with specimens attached was used for further TEM characterizations.

**FDTD simulation**. All the optical simulations were performed with the commercial software FDTD Solutions (Lumerical Inc., Canada). The empirical permittivity of Au[61] was fitted with the multi-coefficient model (MCM)[62]. The refractive index of self-assembly 4-NBT monolayer was set as 1 according to other literatures[63,64]. The surrounding medium was set as water which has the refractive index of 1.33. The refined mesh area with a smallest mesh size of 0.1 nm was utilized to cover the whole structure of GERT thus the accuracy of simulation can be guaranteed. The detailed 3D model of the P-GERT was described in the Supplementary Fig. 17.

**The limit of detection for P-GERTs in liquid**. The limit of detection for P-GERTs was determined by analyzing solutions at various concentrations from 10⁻¹¹ to 10⁻¹⁵ M using ×10 (NA = 0.3), 30 mW laser power and 10 s integration time. Then, 120 SERS spectra of homogenous solutions of P-GERTs were collected at various concentrations for data analysis.

**SERS enhancement factor calculation**. A silicon wafer was used for signal calibration during all the Raman measurements. The enhancement factor is a typical indicator for evaluating the performance of SERS nanoprobes. To evaluate the enhancement effect of P-GERTs and compare them to S-GERTs, we performed enhancement factors (EF) calculation of P-GERTs with various Au core-NBT

incubation time from 0.5 to 960 min. First, the SERS intensity and the ordinary Raman intensity need to be normalized to the number of excited molecules, and then the ratio of the two is calculated to obtain EF:[65]

$$EF = \frac{I_{SERS}/N_{SERS}}{I_{Raman}}/N_{Raman}$$

For normal Raman measurements, the concentration of Raman molecule was 1 mM. To calculate the concentration of SERS active molecules on each probe, the surface area of gold nanospheres with a diameter of 20 nm is about 1256 nm$^2$; assuming a 0.2 nm$^2$ footprint for each Raman molecule[47,66], we evaluate the amount of Raman molecules to be 6280 per P-GERT. Note that this is a maximal amount of SERS active molecules. The use of this value in further calculations gives a minimal estimation for EF.

**AFM-correlated single-particle nano-Raman detection and photostability test**. AFM-correlated Nano Raman system (AIST, Horiba) was used for single-particle detection after the sample (1 pM) was dried on a silicon wafer. First, AFM was used to find the monodispersed particles on the silicon wafer, then the laser spot was coupled to the tip of the silicon probe to collect single-particle Raman signal (633 nm laser, ×100 objective, 110 μW and 10 s integration time). Particles with a distance more than 2 μm from each other were selected for testing and excluding inter-particle coupling effect. Thirty-four single P-GERTs at different regions were tested and then the Raman signal intensity of the single particles was counted. In addition, we performed the photostability of Raman signal of single P-GERT by continuous irradiation for 30 min under the same conditions, and then the variation of Raman signal was analyzed.

**Cell culture**. The human lung cancer cell line H1229 was obtained from American Type Culture Collection (ATCC), and all reagents for the cell culture were purchased from Gibco. The cells were cultured in RPMI 1640 medium supplemented with 10% fetal bovine serum, 100 U mL$^{-1}$ penicillin, and 100 mg mL$^{-1}$ streptomycin at 37 °C in a humidified atmosphere containing 5% CO$_2$.

**Cell proliferation assay**. For evaluating the cytotoxicity of P-GERTs on cells, we investigated the effects of P-GERTs on H1299 cells by using CCK-8 assay (Cell Counting Kit-8, Dojindo). The H1299 were seeded on 96-well plates at a density of $3 \times 10^3$ cells per well, and cultured in RPMI 1640 medium at 37 °C in a humidified atmosphere containing 5% CO$_2$. After 12 h, the culture medium was refreshed and P-GERTs were added at final concentrations of 0.25, 0.5, and 1 nM. Each concentration was set at least five parallel wells. Then, the cells were cultured constantly for 3 days. CCK-8 assay was used on H1299 at 4 h, 1, 2, and 3 days according to the manufacturer's instructions. The absorbance was measured spectrophotometrically by using an enzyme-linked immunosorbent assay microplate reader (Synergy 2, Bio-TEK).

**High-speed and multiplexed Raman cell imaging**. First, the cells were allowed to adhere to the quartz bottomed plates for at least 24 h. For ultrafast cell imaging purpose, cells were then incubated with P-GERTs at a final concentration of 0.05 nM for 6 h. For multiplexed cell imaging, cells were incubated with GERTs with different external Raman reporters (4-MBN, 3-FTP, 3,4-DBT or 2-NT) at a final concentration of 0.05 nM for 6 h. After that cells were washed extensively with phosphate buffered saline (PBS) and fixed with 4% paraformaldehyde for 10 min at room temperature. Excess paraformaldehyde was removed by deionized water and the samples were air-dried before SERS measurements. Cell Raman imaging was performed in a step-by-step mapping mode (Xplora INV, Horiba). In this mode, the cell samples do not move during mapping, and laser is scanned point by point across the samples through rotation of the DuoScan mirror, which greatly improves the imaging speed. For ultrafast cell imaging, the image of a whole cell was obtained within 6 s with a laser power of 370 μW and 0.7 ms acquisition time per pixel for 55 × 66 μm$^2$ field of view (50 × 50 pixels, ×60 objective lens, and 785 nm laser). For multiplexed cell imaging, the image of a whole cell was obtained within 43 s with a laser power of 370 μW and 10 ms acquisition time per pixel for 24 × 66 μm$^2$ field of view (30 × 83 pixels, ×60 objective lens, and 785 nm laser).

**High-speed lymph node in situ Raman imaging**. The nude mice (female, 6 weeks) from Shanghai Laboratory Animal Research Center (SLARC, Shanghai, China) were used and cultured under specific pathogen-free (SPF) conditions. All animal experiments were approved by the Animal Care and Use Committee of Shanghai Jiao Tong University. Mice (n = 3) were injected subcutaneously with P-GERTs in saline (25 μL, 0.5 mg/mL) in the paw of the right hind limb. After 24 h, the mice were anesthetized and ultrafast lymph nodes in situ Raman imaging were performed with 785 nm laser, ×10 objective lens, and 370 μW laser power. In order to minimize the time required for large-area sample imaging, we chose Macro mapping mode for lymph node imaging. During Macro mapping mode, the laser beam is raster-scanned to record an average spectrum across a big area, and the sample is moved by the motorized stage with a step matched to the scanned area size to cover the whole surface. In our experiment, the scanning area was 3.2 × 2.8 cm$^2$, and the motorized stage is moved with a step of 600 μm and raster-scanning by laser beam is performed at the same time to ensure that All Raman signals in the area of 600 × 600 μm$^2$ are collected. The acquisition time can be shortened to 0.7 ms per pixel, and the whole Raman image can be obtained within 52 s (20 × 20 pixels). It is worth noting that, each pixel contains the information over a large area of 600 × 600 μm$^2$, which is tens of thousands of times larger than the normal laser spot area. This allows us to collect sample information as much as possible with short imaging time and few pixels. All the Raman reconstruction images were generated with the characteristic band of P-GERTs at 1340 cm$^{-1}$ by using the LabSpec 6 software.

**Reporting summary**. Further information on research design is available in the Nature Research Reporting Summary linked to this article.

## Data availability
Data are available within the article and supplementary files. The source data underlying Fig. 2a, d, and Supplementary Figs. 2c and 4 are provided as a Source Data file. All other data that support the findings of the study are available from the corresponding author upon reasonable request.

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

## Acknowledgements

We acknowledge the financial support from National Natural Science Foundation of China (Nos. 81571763, 81622026, 61425006, and 81871401), Shanghai Jiao Tong University (Nos. YG2016MS51 and YG2017MS54), the State Key Laboratory of Oncogenes and Related Genes (No. 91-17-28), Shanghai Key Laboratory of Gynecologic Oncology, Guangci Professorship Program of Ruijin Hospital, and Innovation Research Plan supported by Shanghai Municipal Education Commission (No. ZXWF082101).

## Author contributions

Y.Q.Z. and J.Y. conceived the idea for this work and designed the experiments. Y.Q.Z. and Y.Q.G. performed the experiments. J.H. contributed to the numerical calculations. Y.Q.Z., Y.Q.G., B.D.T. and J.Y. wrote and revised the paper. All authors discussed the results and commented on the paper.

## Additional information

**Competing interests:** The authors declare no competing interests.

