## [Peer Review File · Nature Communications]

Reviewers' comments:

Reviewer #1 (Remarks to the Author):

The authors developed new SERS tags for bio-imaging and showed that the SERS imaging speed of a lymph node can be hardly reduced compared to previous studies. More than 50% of their manuscript is about the characterization of a new SERS substrate (which I found it boring and not novel). However, the last part of the manuscript, is remarkably interesting and valuable. I would recommend shortening the manuscript, moving a big part of the first sections into the supplementary information and focusing it into the novel results of the last part. It is hard to follow without losing motivation in its current version.

The section regarding the growing of the 2G tags is highly speculative.

Page 6 and figure 2. The authors should consider the fact that in the 2G structures the surface area is much higher than in the 1G ones and thus the number of molecules contributing to the SERS signal. This number should be carefully evaluated when comparing the 2 orders of magnitude difference in the enhancements (figure 2c). This plus the higher Raman cross-section of 1-NBT could explain the differences observed, without too much contribution of the hot-spots.

Reviewer #2 (Remarks to the Author):

In this manuscript, the authors reported the second-generation gap-enhanced Raman tags (2G GERTs) with multiple hot spots. To form the plasmonic Au nanoparticles containing an interior nanogap with a petal-like shell structure, the authors employed 4-nitrobenzenethiol (4-NBT) instead of 1,4-benzenedithiol (1,4-BDT) based on wet chemistry-based nanoparticle synthesis. The formation of an interior nanogap and the morphology of shell surface were systemically controlled by tuning the incubation time of 4-NBT on the Au core, and the far/near-field plasmonic properties of 2G GERTs were investigated via UV-vis extinction spectra, Raman spectra, and finite-difference time-domain (FDTD) calculations. The authors claim that the Raman signal intensity was highly enhanced compared to other intra-nanogap structures due to multiple hot spots on the external petal-like shell structure, and the detection sensitivity of 2G GERTs was down to ~ 1 fM. In addition, the authors showed capability of 2G GERTs for the high-speed and high-contrast wide-area multiplexed Raman imaging of cell and tissue. Although there are some very nice and promising results, the authors are missing several key references, and there are overclaims and misclaims. Further, the main advance and major point of this paper on interior nanogap particles are not clear. The definition of 2G GERTs is also not well described – why and how this structure can be the next-generation nanogap particle in what aspects? The main aspect of intranagap particles is signal reproducibility and quantitative nature – how this 2nd generation particles improve over the previous version? It seems that the particle the authors report should generate less reproducible results over the previous version due to random and poorly controllable petal structures and their coupling distances. It is also important to note that the structures are somewhat similar to the previous paper (Small, 2016, 12, 4726–4734). Further, additional experimental evidences and in-depth analysis are required to accurately validate the results of this study and to emphasize the advantages and the scientific interpretation of the improvement in the SERS properties more clearly. Based on these, this reviewer cannot recommend publication of this work in Nature Communications. This paper can be probably submitted to a more specialized journal, such as Advanced Functional Materials and Small, after clearly addressing the following comments.

1. Other important and relevant references should be cited, discussed and compared to the authors' paper: Small 2016, 12, 4726–4734; ACS Nano 2014, 8, 6372–6381; J. Am. Chem. Soc.

2014, 136, 16317–16325; Adv. Mater. 2013, 25, 1022–1027; ACS Cent. Sci. 2018, 4, 277–287; ACS Nano 2016, 10, 11066–11075; J. Am. Chem. Soc. 2014, 136, 6838–6841; Anal. Chem. 2016, 88, 7828–7836; J. Phys. Chem. C 2016, 120, 15385–15394.

2. When considering the surface area of the Au core inside the shell (i.e., 22 nm in diameter) and that of petal-like shell structure, the surface area of the latter will be much larger. This means that the total number of attached Raman molecules are obviously higher, in case of 2G GERTs after adsorption of reporter molecules on the external surface of the shells. Despite this, the Raman intensity of 2G GERTs was enhanced ONLY 47% compared to that of 2G GERTs before external surface decoration (Figure 2a). If the Raman intensity is normalized by the total number of Raman reporters on a particle, 2G GERTs with additional Raman reporters on the petal-like shell surface may show a much weaker Raman signal enhancement factor.

3. In Figure 3c, it is unclear whether the change of Raman signal intensity with incubation time of Raman reporter on Au core depends mainly on the number of Raman molecules in the Au core or the morphology of the eventually formed nanoparticles. Which incubation time maximizes the number of Raman reporters on the Au core? What is the exact reason why the Raman signal intensity decreases as the incubation time increases from 5 to 960 min even with the fact that all particles show a petal-like shell structure?

4. What do the author want to mainly emphasize in this study? The signal enhancement effect in the interior nanogap due to the formation of the petal-like shell structure? Or the 'second-generation' signal enhancement based on the formation of the 'additional' multiple hot spots on the external petal-like shell structures? This should be clearly addressed with the additional references and supporting evidences that should be added.

5. In Raman signal-based cell imaging, how were 2G GERTs selectively and specifically located only in the H1229 target cells without cell-targeting ligands on the surface of 2G GERTs? The cell imaging results should be obtained in a larger area with more clear images to clarify the target-cell-selective localization of 2G GERTs. The mechanism with the supporting evidences should be provided.

6. Throughout the manuscript, the detailed description of the FDTD calculation process is not presented. Please provide the specific conditions of FDTD calculation to the 'Methods' section. In particular, please show how the REAL structure of 2G GERTs with very complex surface morphology (Figure 1b and c) was modeled in FDTD calculations.

7. What is the meaning of 'Percentage' in Figure 5e? It seems that the label of y-axis is incorrectly written — it should be 'Raman intensity at 1340 cm⁻¹' instead of 'Percentage'. Please check this point again and correct this.

Reviewer #3 (Remarks to the Author):

Second-generation gap-enhanced Raman tags with multiple "hot spots" and their performance in SERS bioimaging is described in this manuscript. The synthesis, the encoding mechanism and the SERS response is assessed by authors along with cellular and tissue SERS imaging. The production of high sensitive and reproducible SERS tags together with advances in imaging techniques is highly important to improve imaging resolution and speed. The findings are very interesting and the experiments well supported however there are several important aspects that are lacking from the current manuscript, and it would greatly add to the strength of this work if the authors could provide more data and address the following points:

- Related with the synthetic procedure, the petal-like structure formation is not clear to me. It would be helpful to include numbers related to the amount of 4-NBT molecules/nm² available. Did the authors try to add an excess of 4-NBT to the solution? Are the incubation times independent to the amount of 4-NBT? As thiol groups have high affinity to Au surface, higher adsorbance of the molecule is expected when an excess of the molecule is present.

- In order to understand better the interior nanogap evolution, which even the authors say is unclear (Fig 3Viii) after long incubation times using 4-NBT, I recommend to perform electron tomography of the particles.

- Stability studies of the 2G GERTs are missing.

- It is known that for deep-tissue optical imaging the use of longer wavelengths (further NIR range) becomes favourable for in vivo imaging thanks to the "Biological transparency windows." The best SERS values in this work are shown using a 638 nm laser excitation; however for cell and in vivo imaging a 785 nm laser is normally used. The study should be consistent, therefore, data related with single-NP detection at 785 nm which is used later for in vivo imaging purposes, should be included as well. The limit of detection using these particles at 785 nm should be added.

- Laser information should be included as 638 nm is not common laser.

- For in vitro imaging an important point is the stability of the signal over long periods of time. I recommend authors to perform in vitro imaging of live cells over extended periods of time to prove the stability of the signal.

- In Figure 6 b-d what is the size of imaged area? How many seconds are needed to scan a higher area section (for example 0.5 mm²)?

- For in vivo imaging depth penetration is important. Did the authors perform any depth penetration study using the SERS tags? Lymph nodes are relative close to the skin; has any other deeper experiment been done?

To summarize, I think that the presented work is of interest but the mentioned points should be clarified to merit publishing in Nature communication.

Response to reviewers' comments

Reviewer 1

Comment 1: The authors developed new SERS tags for bio-imaging and showed that the SERS imaging speed of a lymph node can be hardly reduced compared to previous studies. More than 50% of their manuscript is about the characterization of a new SERS substrate (which I found it boring and not novel). However, the last part of the manuscript, is remarkably interesting and valuable. I would recommend shortening the manuscript, moving a big part of the first sections into the supplementary information and focusing it into the novel results of the last part. It is hard to follow without losing motivation in its current version.

Response: Thanks so much for the suggestions from the reviewer. According to the reviewer's recommendation, we have moved Figure 2 and part of Figure 3 to the supplementary information and also shorten the manuscript for focusing more on high-speed bioimaging part. The title, abstract, and introduction part have also been revised accordingly for emphasizing the motivation of this manuscript, namely, ultrabright gap-enhanced Raman tags for high-speed bioimaging. The Figure 2 and a part of Figure 3 in the previous version and the corresponding explanation part in the Results have been moved into the Supplementary Information, and whole characterization part has been shortened for more focusing on the imaging part.

Title: Second-generation gap-enhanced Raman tags with multiple "hot spots" → Ultrabright gap-enhanced Raman tags for high-speed bioimaging

Abstract: Surface-enhanced Raman spectroscopy (SERS) is advantageous over fluorescence for bioimaging due to ultra-narrow linewidth of the fingerprint spectrum and weak photo-bleaching effect. However, the existing SERS imaging speed lags far behind practical needs, mainly limited by Raman signals of SERS nanoprobe. In this work, we report ultrabright gap-enhanced Raman tags (GERTs) with strong electromagnetic hot spots from interior sub-nanometer gaps and external petal-like shell structures, larger immobilization surface area, and Raman cross section of reporter molecules. These GERTs show two orders of magnitude stronger signals compared to those with smooth external shell reported previously, reaching a Raman enhancement factor beyond 5×10^9 and a detection sensitivity down to a single-nanoparticle level. We use a 370 μW laser to realize high-resolution cell imaging (2500 pixels for $55 \times 66 \mu\text{m}^2$) within 6 s and high-contrast (a

signal-to-background ratio of 80) wide-area ($3.2 \times 2.8 \text{ cm}^2$) sentinel lymph node imaging within 52 s. These nanoprobe offer a potential solution to overcome the current bottleneck in the field of SERS-based bioimaging.

The Figure 2 and a part of Figure 3 in the previous version and the below paragraph have been moved to the Supplementary Information.

“We investigated the far- and near-field optical properties of P-GERTs with different morphologies. The extinction spectra of P-GERTs with eight different incubation time are shown in Supplementary Figure 6a, all exhibiting only one plasmon resonance peak in the visible range. As the incubation time is increased from 0.5 to 5 min, the resonance peak gradually redshifts from 584 to 610 nm, accompanied by a significant broadening of the resonance linewidth. However, as the incubation time is further increased from 5 to 960 min, the resonance peak gradually blueshifts from 610 to 574 nm, accompanied by a slight linewidth narrowing. The near-field optical properties of P-GERTs with varying incubation time are investigated by measuring the averaged SERS signals. We firstly investigate the Raman enhancement contribution from the internal near-field hot spots with the influence by the Au shell morphology induced by the incubation time. The Raman spectra and the band intensities at 1340 cm^{-1} of P-GERTs (without the external decoration of 4-NBT) with different incubation time for the excitation of 638 and 785 nm were plotted for comparison in Supplementary Figure 6b. They all exhibit a similar trend that the Raman intensity increases at the initial stage and reaches the maximum when the incubation time is 5 min but then decreases when the incubation was further prolonged until to 960 min. Both this phenomenon and the aforementioned far-field extinction spectral evolution can be explained by the morphology changes of P-GERTs. The morphology changes of P-GERTs may related to the adsorption amount of 4-NBT molecules on Au cores and the poor affinity between Au and the nitro group. For the reason that the Raman signal intensity decreases as the incubation time increases from 5 to 960 min, we presume that the internal nanogaps become more incomplete during this process, which also leads to the much weaker Raman enhancement.”

Comment 2: The section regarding the growing of the 2G tags is highly speculative.

Response: Thank you for your comments. In order to study the growth process of 2G GERTs more accurately and to prove our conjecture, we have added time-lapse TEM analysis results (see Figure 2c-d and below Figure R1) and the relevant description in the revision as follows.

Page 9-10: “To understand how the number of 4-NBT molecules on Au cores influences the growth of Au shells, we investigated their growth process using time-lapse TEM analysis. Snapshots of intermediate products were obtained by terminating the Au shell growth process at different time points (from 1 to 12 min) after 4-NBT modified Au cores were introduced into the reaction. When the incubation time is 0.5 and 2 min, 4-NBT exhibits an adsorption density of around 2.7 and 3.3 molecules/nm² on Au cores, according to its molecular footprint of ~0.2 nm² and the plasmon resonance shifts in Supplementary Figure 3, and they primarily lie down on the Au core surface and form a leaky monolayer (Fig. 2b). The Au shell growth starts with a relatively large nucleation on the core, and gradually turns outward to form the shell and wrap around the Au core. Since the 4-NBT molecules are in lying-down orientation and the nitro group has a small contact area with the Au shell, the Au shell can easily form a complete piece which is similar to S-GERTs. Further increasing of incubation time to 5 – 960 min allows the improvement of the adsorption density to 4 – 5 molecules/nm², and 4-NBT molecules form a more densely packed monolayer on the cores in a more vertical state (Fig. 2c). Due to the poor affinity between Au and the nitro group, the Au shell can only grow from a number of small nucleation sites on cores, and gradually forms petal-like structures, which is more conducive to light penetration. In addition, the internal nanogaps also become more discontinuous. Time-lapse TEM examination offers us a preliminary understanding of the petal-like shell growth mechanism, but a more advanced characterization such as high-resolution in situ TEM may further disclose more insights.”

Figure R1. TEM snapshots of Au shell formation during the initial 12 min reaction and their corresponding schematic growth process for two routes of Au cores decorated by 4-NBT molecules for (a) 0.5-2 min and (b) 5-960 min.

Comment 3: Page 6 and Figure 2. The authors should consider the fact that in the 2G structures the surface area is much higher than in the 1G ones and thus the number of molecules contributing to the SERS signal. This number should be carefully evaluated when comparing the 2 orders of magnitude difference in the enhancements (figure 2c). This plus the higher Raman cross-section of 1-NBT could explain the differences observed, without too much contribution of the hot-spots.

Response: Thanks for reviewer's comments. First, we want to clarify that in this work 2G GERTs have two orders of magnitude higher than 1G GERTs in **total Raman signal intensity (instead of the Raman enhancement factor)** with the same particle concentration and measurement conditions. Namely, this means the Raman signal of a single 2G GERT is two orders of magnitude brighter than a single 1G GERT. We have revised the statement in the revision to avoid the misunderstanding. Second, we totally agree with the points of the reviewer that the surface area and the number of reporter molecules in 2G structures are higher than that in 1G ones, in addition to the higher molecular Raman cross-section. Therefore, in the revision we try to avoid overemphasizing the contribution of hotspots from the external surface. For example, we remove "multiple hot spots" from the title and rewrite the abstract. In addition, for further understanding, we carefully evaluated the Raman enhancement factor from the external hotspots due to the petal-like structures. We estimated that the number of molecules on the surface area of 2G structures is about thirty times the number of molecules in the internal hotspots. We have shown (in below Figure R2) that after decorating 4-NBT molecules in the external nanogaps, the Raman

intensity improved 47% and 21% for 638 and 785 nm excitation (when the incubation time is 10 min), respectively. Thus, we can estimate the enhancement factor of the external hotspots is about 1×10^8 and 2.67×10^7 for 638 and 785 nm excitation, respectively. These enhancement factors are both roughly one order of magnitude smaller than that of internal hotspots. This again confirms the dominant contribution from the internal hotspots.

Therefore, we have revised the statements in the revision to avoid overemphasizing the contribution of external hotspots in below parts.

Title: removing “multiple hot spots”

Abstract: “ ... In this work, we report ultrabright gap-enhanced Raman tags (GERTs) with strong electromagnetic hot spots from interior sub-nanometer gaps and external petal-like shell structures, larger immobilization surface area, and Raman cross section of reporter molecules...”

Page 6:“ ...Second, the larger surface area and multiple external nanogaps in P-GERTs offer a larger number of immobilization site and hot spots for reporter molecules, therefore achieving a strong Raman enhancement...”

Page 11:“... By considering a larger surface area of external shell and a larger molecular Raman cross section, we have also found that the Raman enhancement factor from the external hot spots is roughly one order of magnitude small than that from the internal hot spots in P-GERTs although the overall Raman signal is dramatically enlarged. This is probably explained by the difficulty of molecular diffusion into the external nanogaps during the self-assembly process.”

Figure R2. Comparison of Raman band intensity (1340 cm^{-1} for P-GERTs and 1055 cm^{-1} for S-GERTs) of P-GERTs and S-GERTs before and after adsorption of reporter molecules on the external surface of the shells excited by 532, 638 and 785 nm laser.

Reviewer 2

In this manuscript, the authors reported the second-generation gap-enhanced Raman tags (2G GERTs) with multiple hot spots. To form the plasmonic Au nanoparticles containing an interior nanogap with a petal-like shell structure, the authors employed 4-nitrobenzenethiol (4-NBT) instead of 1,4-benzenedithiol (1,4-BDT) based on wet chemistry-based nanoparticle synthesis. The formation of an interior nanogap and the morphology of shell surface were systemically controlled by tuning the incubation time of 4-NBT on the Au core, and the far/near-field plasmonic properties of 2G GERTs were investigated via UV-vis extinction spectra, Raman spectra, and finite-difference time-domain (FDTD) calculations. The authors claim that the Raman signal intensity was highly enhanced compared to other intra-nanogap structures due to multiple hot spots on the external petal-like shell structure, and the detection sensitivity of 2G GERTs was down to ~ 1 fM. In addition, the authors showed capability of 2G GERTs for the high-speed and high-contrast wide-area multiplexed Raman imaging of cell and tissue. Although there are some very nice and promising results, the authors are missing several key references, and there are overclaims and misclaims. Further, the main advance and major point of this paper on interior nanogap particles are not clear. The definition of 2G GERTs is also not well described – why and how this structure can be the next-generation nanogap particle in what aspects? The main aspect of intrananogap particles is signal reproducibility and quantitative nature – how this 2nd generation particles improve over the previous version? It seems that the particle the authors report should generate less reproducible results over the previous version due to random and poorly controllable petal structures and their coupling distances. It is also important to note that the structures are somewhat similar to the previous paper (Small, 2016, 12, 4726–4734).

Further, additional experimental evidences and in-depth analysis are required to accurately validate the results of this study and to emphasize the advantages and the scientific interpretation of the improvement in the SERS properties more clearly. Based on these, this reviewer cannot recommend publication of this work in Nature Communications. This paper can be probably submitted to a more specialized journal, such as Advanced Functional Materials and Small, after clearly addressing the following comments.

Comment 1: The main advance and major point of this paper on interior nanogap particles are not clear.

Response: Thanks for pointing out the unclear statement of the main advance and the major point of our manuscript. **This work is mainly emphasizing the ultrahigh brightness of a new version of GERTs, which reaches a single-nanoparticle detection sensitivity and allows us to perform**

high-speed SERS imaging, instead of only emphasizing the role of interior nanogaps. The existing SERS imaging speed lags far behind practical needs, mainly limited by overall Raman signals of SERS tags. In this work, we utilized three factors to improve the total brightness of GERTs. First, the internal nanogap still exists in the new version of GERTs, and the plasmon coupling between the Au core and shell produces strong electromagnetic enhancement in internal nanogap region. The petal-like shell structure of 2G GERTs is more conducive to light penetration than the complete shell of 1G GERTs, resulting in more efficient light excitation for the internal nanogaps. Second, the larger surface area and multiple external nanogaps in 2G GERTs offer a larger number of immobilization site and hot spots for reporter molecules, therefore achieving a strong Raman enhancement. Third, 4-NBT molecules have a larger Raman cross section than 1,4-BDT due to the intrinsic chemical structure.

Therefore, the title, abstract, and introduction part have also been significantly revised for emphasizing the main advance and the major point of this manuscript: ultrabright gap-enhanced Raman tags for high-speed bioimaging.

For example, Introduction part: "...SERS tags have shown great prospects for in vivo bioimaging applications such as bright optical guidance for intraoperative detection of cancer margins, residual cancerous foci, microscopic or disseminated tumors, and sentinel or metastasized lymph node. However, the existing SERS imaging speed lags far behind clinical needs. For example, it typically takes hours to acquire a wide-area Raman in vivo image. The current bottlenecks for high-speed SERS bioimaging mainly include the overall Raman signals (instead of the Raman enhancement factor) of SERS tags, imaging method of Raman system, and the data process speed..."

Comment 2: The definition of 2G GERTs is also not well described – why and how this structure can be the next-generation nanogap particle in what aspects?

Response: Thanks for pointing out the unclear definition of 2G GERTs. In the revision, we remove the definition of second-generation of GERTs, and call them as GERTs with petal-like shells (P-GERTs). Those 1G GERTs with smooth-surface shells are called S-GERTs. Mainly two different things can be observed between P-GERTs and S-GERTs: (1) particle morphology and (2) overall Raman brightness. P-GERTs have two orders of magnitude improvement in Raman signal strength (instead of enhancement factor) due to the petal-like shell structures when compared to the S-GERTs with the same particle concentration and measurement conditions. We mainly focus on the improvement of Raman brightness of particles in this work, by combining the enhancement effect from internal and external nanogaps hot spots. In the case that the size of P-GERTs is almost unchanged when compared with S-GERTs, surface area is largely increased due to the petal-like

structure and thus more molecules contribute to the SERS signal. At the same time, more electromagnetic hotspots are introduced through the change of the particle morphology, so that the overall signal of P-GERTs is greatly enhanced and can be used for high-speed imaging.

Comment 3: The main aspect of intra-nanogap particles is signal reproducibility and quantitative nature, how this 2nd generation particles improve over the previous version? It seems that the particle the authors report should generate less reproducible results over the previous version due to random and poorly controllable petal structures and their coupling distances. It is also important to note that the structures are somewhat similar to the previous paper (Small, 2016, 12, 4726–4734).

Response: Thanks again. The nanogap hotspots of P-GERTs (previously called 2G GERTs) are not limited to intra-nanogap, but also include contributions from external particle morphology. It has been found that the main improvement of P-GERTs is two orders of magnitude improvement in Raman signal strength, instead of signal reproducibility and quantitative nature, when compared to the S-GERTs (previously called 1G GERTs).

Indeed, we performed the reproducibility test of P-GERTs on the single-particle level using AFM-correlated nano-Raman measurement. The results (as shown in below Figure R3 and Figure 4d in the old version) show that more than 80% of Raman signals produced from single nanoparticle measurements are between 350 and 450 counts, demonstrating a still quite good uniformity at the single-particle level. Although the petal-like Au shell structures of P-GERTs are random and poorly controlled compared with the completely smooth Au shell of S-GERTs, each P-GERTs has a very large number of petal-like structures on the surface, and the final signal intensity of each nanoparticle shows an average result with a still acceptable uniformity. Due to too low signal, we did not manage to obtain the signal of single particle of S-GERTs. Therefore, we cannot do the direct comparison. But we found a very similar nanoparticle structure in previous work (Nature Nanotechnology, 2011, 6, 452-460), which reported a kind of highly uniform and reproducible SERS tags: most EF values (>90%) of single particle are narrowly distributed in the range from 1.0×10^8 to 5.0×10^9 . Compared to these tags, our P-GERTs indeed show a slightly worse uniformity.

Figure R3. Distribution diagram of the SERS intensity at 1340 cm⁻¹ of thirty-four single mesoporous silica coated P-GERTs from different areas on silicon wafer.

It has been noticed that the previous paper (Small, 2016, 12, 4726–4734) presented an excellent work. **However, the previous paper mainly focused on the effect of surface roughness of particle on the electromagnetic field inside the interior nanogap and the Raman signals. In our work, we focused on the improvement of overall signals of SERS tags by utilization of both internal and external nanogap hotspots, large surface area, and molecular Raman cross section, since we realize that the overall Raman signal is the bottleneck of SERS imaging speed.** In addition, the Au shell geometry and its formation of P-GERTs are different, as well as the formation of internal nanogap, when compared with previous paper (Small, 2016, 12, 4726–4734, also shown in below Figure R4). The Au shell of P-GERTs is petal-like structure and composed of many pieces, while the shell of Au-RNNPs in previous work is in a whole piece with rough surface. The internal gaps of P-GERTs are formed by the embedded 4-NBT molecules, while the internal gaps of Au-RNNPs in previous work are formed by DNA molecules. Moreover, not every kind of molecule can form the petal-like shell structures of P-GERTs. We have tried many kinds of molecules, and only 4-NBT can form this structure so far. The preliminary proposed growth mechanism is shown in the above Figure R1.

Therefore, according to the reviewer’s comments, we have added the paper (Small, 2016, 12, 4726–4734) in the reference list when we explain the shell growth mechanism (page 10) and added the corresponding discussion as follows.

Page 15: “...More than 80% of 34 single-NP measurements produced Raman signals between 350 and 450 counts, demonstrating a quite good uniformity at the single-NP level (Fig. 4d). Although the petal-like structures of P-GERTs are random and poorly controlled, each P-GERT has a large

number of petal-like structures on the surface, which results in an average Raman signal of each NP in a decent uniformity...”

Figure R4. a, Schematic representation of rough Au shell formation on DNA-AuNPs with a different amount of hydroxylamine as a reducing agent. b, TEM images of Au-RNNPs with 40 nm DNA-AuNPs. The ratio (HA/Au^{3+}) of 2 was used for b1 and b4, the ratio of 5 was used for b2 and b5, and the ratio of 30 was used for b3 and b6. All the scale bars are 20 nm. (Copied from Small, 2016, 12, 4726–4734)

Comment 4: Other important and relevant references should be cited, discussed and compared to the authors’ paper: Small 2016, 12, 4726–4734; ACS Nano 2014, 8, 6372–6381; J. Am. Chem. Soc. 2014, 136, 16317–16325; Adv. Mater. 2013, 25, 1022–1027; ACS Cent. Sci. 2018, 4, 277–287; ACS Nano 2016, 10, 11066–11075; J. Am. Chem. Soc. 2014, 136, 6838–6841; Anal. Chem. 2016, 88, 7828–7836; J. Phys. Chem. C 2016, 120, 15385–15394.

Response: Thanks for suggestion of these relevant excellent papers. All these articles were mainly focusing on the investigation and optimization of nanoparticles with interior nanogaps and their biomedical applications including the bioimaging and theranostics. They provide us the indication of potential biomedical applications for P-GERTs. We have added these papers in the reference list and the corresponding discussion in the introduction part on page 3.

Page 3: ...“Compared to the nanogaps randomly formed in NP aggregates, S-GERTs provide strong, uniform and stable SERS signals by embedding Raman molecules in the interior nanogap “hot spots”.^{10,11} More efforts have been recently spent to understand and further improve the performance of these NPs.²⁹⁻³⁷ However, the overall Raman signal of these S-GERTs is still not strong enough to reach high-speed imaging...”

Comment 5: When considering the surface area of the Au core inside the shell (i.e., 22 nm in diameter) and that of petal-like shell structure, the surface area of the latter will be much larger. This means that the total number of attached Raman molecules are obviously higher, in case of 2G GERTs after adsorption of reporter molecules on the external surface of the shells. Despite this, the Raman intensity of 2G GERTs was enhanced ONLY 47% compared to that of 2G GERTs before external surface decoration (Figure 2a). If the Raman intensity is normalized by the total number of Raman reporters on a particle, 2G GERTs with additional Raman reporters on the petal-like shell surface may show a much weaker Raman signal enhancement factor.

Response: Thank the reviewer for raising this concern. We agree that the Raman enhancement factor from the external hot spots will be much weaker than that from the internal hot spots when normalized by the total number of Raman reporters on P-GERTs (previously called 2G GERTs). In this work, what we want to emphasize is the overall ultra-high brightness of P-GERTs (which allows for high-speed bioimaging), instead of the enhancement factor of P-GERTs. We take advantage of all the favorable factors, such as the internal and multiple external nanogaps, large surface area, large 4-NBT molecule numbers, and larger Raman cross sections of molecules, to achieve the ultra-high SERS intensity of P-GERT as shown in Fig.4. Please see more discussion in the **Response to Comment 3** of the **Reviewer 1**.

Comment 6: In Figure 3c, it is unclear whether the change of Raman signal intensity with incubation time of Raman reporter on Au core depends mainly on the number of Raman molecules in the Au core or the morphology of the eventually formed nanoparticles. Which incubation time maximizes the number of Raman reporters on the Au core? What is the exact reason why the Raman signal intensity decreases as the incubation time increases from 5 to 960 min even with the fact that all particles show a petal-like shell structure?

Response: Thanks for your comments. The change of Raman signal intensity with incubation time of Raman reporter on Au core depends mainly on the morphology of the eventually formed nanoparticles. The number of Raman molecules on Au cores will increase as the adsorption time increases, however we found that the Raman intensity of 2G GERTs first increases and reaches the maximum when the incubation time of Au core-NBT is 5 min but then decreases when the

incubation was further prolonged until to 960 min. Thus the Raman intensity of 2G GERTs depends mainly on the morphology of nanoparticles rather than the number of Raman molecules on Au cores.

For the reason that the Raman signal intensity decreases as the incubation time increases from 5 to 960 min, we presume that the internal nanogaps become more incomplete during this process, which also leads to the much weaker Raman enhancement.

Therefore, we have added some relevant descriptions in the Supplementary Information as follows.

“We investigated the far- and near-field optical properties of P-GERTs with different morphologies. The extinction spectra of P-GERTs with eight different incubation time are shown in Supplementary Figure 6a, all exhibiting only one plasmon resonance peak in the visible range. As the incubation time is increased from 0.5 to 5 min, the resonance peak gradually redshifts from 584 to 610 nm, accompanied by a significant broadening of the resonance linewidth. However, as the incubation time is further increased from 5 to 960 min, the resonance peak gradually blueshifts from 610 to 574 nm, accompanied by a slight linewidth narrowing. The near-field optical properties of P-GERTs with varying incubation time are investigated by measuring the averaged SERS signals. We firstly investigate the Raman enhancement contribution from the internal near-field hot spots with the influence by the Au shell morphology induced by the incubation time. The Raman spectra and the band intensities at 1340 cm^{-1} of P-GERTs (without the external decoration of 4-NBT) with different incubation time for the excitation of 638 and 785 nm were plotted for comparison in Supplementary Figure 6b. They all exhibit a similar trend that the Raman intensity increases at the initial stage and reaches the maximum when the incubation time is 5 min but then decreases when the incubation was further prolonged until to 960 min. Both this phenomenon and the aforementioned far-field extinction spectral evolution can be explained by the morphology changes of P-GERTs. The morphology changes of P-GERTs may related to the adsorption amount of 4-NBT molecules on Au cores and the poor affinity between Au and the nitro group. For the reason that the Raman signal intensity decreases as the incubation time increases from 5 to 960 min, we presume that the internal nanogaps become more incomplete during this process, which also leads to the much weaker Raman enhancement.”

Comment 7: What do the author want to mainly emphasize in this study? The signal enhancement effect in the interior nanogap due to the formation of the petal-like shell structure? Or the ‘second-generation’ signal enhancement based on the formation of the ‘additional’ multiple hot spots on the external petal-like shell structures? This should be clearly addressed with the additional references and supporting evidences that should be added.

Response: Sorry for unclear statements in the previous version. What we want to emphasize in this study is the ultrahigh brightness of P-GERTs, which reaches a single-nanoparticle detection sensitivity and allows for high-speed SERS imaging. Please refer to more discussion in the **Response to Comment 1** of the **Reviewer 2**.

Comment 8: In Raman signal-based cell imaging, how were 2G GERTs selectively and specifically located only in the H1229 target cells without cell-targeting ligands on the surface of 2G GERTs? The cell imaging results should be obtained in a larger area with more clear images to clarify the target-cell-selective localization of 2G GERTs. The mechanism with the supporting evidences should be provided.

Response: Sorry for the unclear statement about the cell imaging. In Raman signal-based cell imaging, P-GERTs were passively internalized by H1299 cells without any selective and specific labelling for H1229 cells. H1299 is a kind of common lung cancer cell and is only used as a demo for ultrafast cell imaging in this study. Macropinocytosis might be an underlying mechanism for the uptake of P-GERTS, which has been implicated in the cellular internalization of nanoparticles with a similar composition as the P-GERTs [Sci Transl Med, 2015, 7, 271; Adv Drug Deliver Rev, 2013, 65 (1), 71-79].

Therefore, we have revised the discussion in page 16-17 for a better and clear statement as follows: “...As a demonstration, Figure 5b-d show the bright-field image, Raman image and overlay of a H1299 cell passively stained with the P-GERTs...”

In addition, we have added large-area cell images containing a number of cells in Supplementary Figure 11 (also shown in below Figure R5) as required, and revised manuscript on page 17 as follows: “...Raman imaging of multiple cells in a larger area (61×41 pixels for $380 \times 240 \mu\text{m}^2$ field of view) can also be accomplished within 7 s (Supplementary Figure 11). Such high imaging speed can be attributed to the significant improvements in SERS tag signal, imaging system, and data processing method...”

Figure R5. Large-area SERS images containing multiple H1299 cells. The top Raman image ($110 \times 140 \mu\text{m}^2$, 50×50 pixels) was obtained within 6 s, and the bottom Raman image ($380 \times 240 \mu\text{m}^2$, 61×41 pixels) was obtained within 7 s. All scale bars are $50 \mu\text{m}$. The Raman images are plotted using the Raman band (1340 cm^{-1}) of 4-NBT.

Comment 9: Throughout the manuscript, the detailed description of the FDTD calculation process is not presented. Please provide the specific conditions of FDTD calculation to the ‘Methods’ section. In particular, please show how the REAL structure of 2G GERTs with very complex surface morphology (Figure 1b and c) was modeled in FDTD calculations.

Response: Sorry for our mistake. According to the reviewer’s comments, we have added below detailed description of the FDTD calculation process to the *Methods* section (see page 23-24).

Page 23-24: “**FDTD simulation.** All the optical simulations were performed with the commercial software FDTD Solutions (Lumerical Inc., Canada). The empirical permittivity of Au^{59} was fitted with the multi-coefficient model (MCM).⁶⁰ The refractive index of self-assembly 4-NBT monolayer was set as 1 according to other literatures.^{61,62} The surrounding medium was set as water which has the refractive index of 1.33. The refined mesh area with a smallest mesh size of 0.1 nm was utilized to cover the whole structure of GERT thus the accuracy of simulation can be guaranteed. The detailed 3D model of the P-GERT was described in the Supplementary Figure 15”.

The real structure of P-GERTs is very complex with random petal-like morphology, and in simulations we used a 3D model as indicated in below Figure R6. The below figure and the corresponding description of the 3D model have been added into the Supplementary Information.

Figure R6. The 3D model of a P-GERT used in FDTD simulation.

The real structure of P-GERTs is very complex with random petal-like shell structures and therefore we used some small Au nanospheres on the surface to mimic the petal-like structures. The P-GERT has an inner Au core of 30 nm in diameter, a gap of 1 nm in thickness, and a Au shell of 8 nm in thickness. The Au nanospheres on the surface are 18 nm in diameter and the total particle size is 70 nm. The total eighteen Au nanospheres (indicated by yellow color and with alphabetic letters, on the right of above Figure R6) were evenly distributed on three perpendicular circumferences of the particle along the X, Y, and Z axis, with eight nanospheres distributed on each direction. A 2 nm nanogap between the nanospheres is formed on the surface, which mimics the external nanogaps on the P-GERTs surface. Next, eight Au nanospheres (indicated by red color and with numbers, on the right of above Figure R6) are placed in the center of each rest blank areas. The distance between the center of red Au nanospheres and the center of the inner core is 26 nm. The red Au nanospheres may overlap with neighbor yellow Au nanospheres, which also tries to mimic the random petal-like structures.

Comment 10: What is the meaning of ‘Percentage’ in Figure 5e? It seems that the label of y-axis is incorrectly written — it should be ‘Raman intensity at 1340 cm^{-1} ’ instead of ‘Percentage’. Please check this point again and correct this.

Response: We apologize for our mistakes. The label of x-axis should be ‘Raman intensity at 1340 cm^{-1} ’, and the label of y-axis should be ‘Percentage’. Percentage means the population distribution of the SERS intensity at 1340 cm^{-1} of thirty-four single mesoporous silica coated P-GERTs from different areas on silicon wafer. **We have corrected this in the revision (see above Figure R3).**

Reviewer #3

Comment 1: Related with the synthetic procedure, the petal-like structure formation is not clear to me. It would be helpful to include numbers related to the amount of 4-NBT molecules/ nm^2

available. Did the authors try to add an excess of 4-NBT to the solution? Are the incubation times independent to the amount of 4-NBT? As thiol groups have high affinity to Au surface, higher adsorbance of the molecule is expected when an excess of the molecule is present.

Response: Sorry for unclear description about the synthesis procedure. Yes, indeed we added an excess of 4-NBT molecules to the Au core solution with a constant concentration of 10 mM. Therefore, adsorption of the number of 4-NBT molecules on Au cores is only dependent on the incubation times.

The 4-NBT molecules were adsorbed on Au cores as a monolayer with a footprint of $\sim 0.2 \text{ nm}^2$ [Phys. Chem. Chem. Phys., 2017, 19, 4478], thus there is about five 4-NBT molecules/ nm^2 . This assumption represents the theoretical maximum number of molecules (in the case of a monolayer). 4-NBT molecules were self-assembled on Au cores via Au-S covalent bonding but cannot form a multilayered molecular structure due to the lack of the second thiol group even when a large excess of 4-NBT was added [Nanoscale, 2017, 9, 2213; Small Methods, 2017, 1, 1600032]. The refractive index changes of the surrounding environment induced by the 4-NBT molecular decoration causes a red shift in the localized surface plasmon resonance (LSPR) of the Au cores. Therefore, we performed real-time monitoring of 4-NBT immobilization on the Au cores by measuring their LSPR spectra. As shown in below Figure R7, the LSPR peak of the Au cores exhibits a continuous red shift during 960 min incubation with 4-NBT. The LSPR shift reaches 2.6 nm within 60 min, and remains 2.6 nm when 960 min incubation. Therefore, we infer that 2.6 nm shift means self-assembled 4-NBT monolayers on Au cores, and there is about five 4-NBT molecules/ nm^2 . The incubation times were 0.5, 2, 5, 10, 20, 30 and 60 min, and the red shift was 1.4, 1.7, 2.1, 2.2, 2.3, 2.5 and 2.6 nm, respectively. Therefore, it can be estimated that there is about 2.7, 3.3, 4, 4.2, 4.4, 4.8 and 5 molecules/ nm^2 when incubation times were 0.5, 2, 5, 10, 20, 30 and 60 min, respectively.

Therefore, we have added below figure into the Supplementary Figure 3 and the corresponding discussion on page 9-10 line.

“...When the incubation time is 0.5 and 2 min, 4-NBT exhibits an adsorption density of around 2.7 and 3.3 molecules/ nm^2 on Au cores, according to its molecular footprint of $\sim 0.2 \text{ nm}^2$ and the plasmon resonance shifts in Supplementary Figure 3, and they primarily lie down on the Au core surface and form a leaky monolayer (Fig. 2b).¹⁴ The Au shell growth starts with a relatively large nucleation on the core, and gradually turns outward to form the shell and wrap around the Au core.⁴³ Since the 4-NBT molecules are in lying-down orientation and the nitro group has a small contact area with the Au shell, the Au shell can easily form a complete piece which is similar to S-GERTs. Further increasing of incubation time to 5 – 960 min allows the improvement of the

adsorption density to 4 – 5 molecules/nm², and 4-NBT molecules form a more densely packed monolayer on the cores in a more vertical state (Fig. 2c)... ”

Figure R7. LSPR shifts of the mixture of Au cores and 4-NBT molecules during 960 min incubation.

Comment 2: In order to understand better the interior nanogap evolution, which even the authors say is unclear (Fig 3Viii) after long incubation times using 4-NBT, I recommend to perform electron tomography of the particles.

Response: Thanks for this great suggestion. We tried our best to look for this characterization technique. However, due to the unavailability of the instrument, we cannot perform electron tomography of the particles now. We have performed more high-resolution TEM imaging and the interior nanogaps can be still identified (as indicated by red arrows in below Figure R8). In addition, we have added time-lapse TEM analysis results (see above Figure R1) to understand better the interior nanogap evolution.

We have added below figure into the Supplementary Information and revised the corresponding description on page 10.

“...Time-lapse TEM examination offers us a preliminary understanding of the petal-like shell growth mechanism, but a more advanced characterization such as high-resolution in situ TEM may further disclose more insights...”

Figure R8. More representative TEM images of P-GERTs when the incubation time of Au cores and 4-NBT molecules is 960 mins. All red arrows indicate the internal nanogaps.

Comment 3: Stability studies of the 2G GERTs are missing.

Response: Thanks for the reviewer's suggestion. We have performed the stability studies of aqueous mesoporous silica (MS)-coated P-GERTs (previously called 2G GERTs) in terms of the extinction spectra and the Raman spectra. As shown in below Figure R9, their extinction and Raman spectra in aqueous samples of different pH values (from 5 to 10) and incubated them with 10% fetal bovine serum (FBS) and saline for 72 h, show quite stable intensity and unchanged spectral profiles. These results indicate excellent materials stability of aqueous P-GERTs suitable for biomedical applications in different environments.

We have added below figure into the Supplementary Information and the corresponding discussion on page 13 as follows.

“...We have additionally performed the stability analysis of aqueous MS P-GERTs. The extinction and Raman spectra in aqueous solution of different pH values (from 5 to 10) and in 10% fetal bovine serum (FBS) and saline for 72 h show quite stable intensity and unchanged spectral profiles (Supplementary Figure 10). These results indicate excellent materials stability of aqueous P-GERTs suitable for biomedical applications in different environments...”

Figure R9. Extinction spectra of MS P-GERTs in (a) aqueous solutions of different pH values in the range of 5–10, (b) 10% FBS and (c) saline for 72 h. Raman spectra of MS P-GERTs in (d) aqueous solutions of different pH values in the range of 5–10, (e) 10% FBS and (f) saline for 72 h.

Comment 4: It is known that for deep-tissue optical imaging the use of longer wavelengths (further NIR range) becomes favourable for in vivo imaging thanks to the “Biological transparency windows.” The best SERS values in this work are shown using a 638 nm laser excitation; however for cell and in vivo imaging a 785 nm laser is normally used. The study should be consistent, therefore, data related with single-NP detection at 785 nm which is used later for in vivo imaging purposes, should be included as well. The limit of detection using these particles at 785 nm should be added.

Response: Thanks for the reviewer’s comments. We have additionally performed the limit of detection and single-NP detection at 785 nm excitation, as indicated in below Figure R10. It can be noticed that the results excited by 785 nm are quite similar to the those by 638 nm, only with a slightly lower (around 70%) Raman intensity compared to those by 638 nm. We also notice that we could constantly detect Raman signals from the MS 2G GERTs for relatively high concentrations, such as 10 pM, 1 pM, and 100 fM, but only signals in about a quarter of measurements for the concentration of 10 fM, and signals in very few cases for the concentration of 1 fM (see Figure R10).

We have added below figure into the Supplementary Information and the corresponding discussion on page 13 as follows.

“...It can be also noticed that the Raman measurements of ultra-low concentration of aqueous P-GERTs excited by 785 nm show quite similar behaviors to those by 638 nm, only with a slightly lower (around 70%) Raman intensity (Supplementary Figure 9)... ”

Figure R10. Single-NP detection of aqueous MS P-GERT solution at 785 nm. (a) Concentration-dependent Raman measurements of aqueous P-GERTs (785 nm laser, 20 mW, 10 s acquisition time, and 60× objective lens). (b) The intensity of Raman band at 1340 cm⁻¹ of 120 measurements from (b) 100, (ii) 10, and (iii) 1 fM P-GERT solution. 100, 10, and 1 fM GERT solutions correspond to 2, 0.2, and 0.02 particle per probing volume, respectively. Blue dotted lines in panel (b-d) represent the background Raman signal (1340 cm⁻¹) without P-GERTs.

Comment 5: Laser information should be included as 638 nm is not common laser.

Response: The 638 nm laser provided by Horiba was an air cooled CLDS laser diode with a linewidth smaller than 0.03 nm.

We have added this information to the *Methods* part on page 23: “...The 638 nm laser provided by Horiba was an air cooled CLDS laser diode with a linewidth smaller than 0.03 nm...”.

Comment 6: For *in vitro* imaging an important point is the stability of the signal over long periods of time. I recommend authors to perform *in vitro* imaging of live cells over extended periods of time to prove the stability of the signal.

Response: Thanks again for the comments. We absolutely agree with the reviewer. Due to the limitations of current experimental conditions, we are unable to perform continuous and long-term live-cell imaging. Instead, we use fixed cells labelled with P-GERTs for the continuous SERS imaging, which may be more suitable for the evaluation of the stability of the Raman signal of pure nanotags, without the interference from the live cells and culture buffer. The results (see Figure S9 and below Figure R11) show super-stable Raman images of a cell during 30-min of continuous irradiation, and therefore proves the excellent photostability of the signal of P-GERTs.

We have added the below figure in the supporting information and the corresponding discussion on page 17-18 as follows.

“...We additionally demonstrated that P-GERTs exhibited excellent long-term photostability during high-speed 30-min continuous cell imaging process (Supplementary Figure 13)...”

Figure R11. Representative super-stable Raman images obtained from a single cell: bright-field image (top left) and Raman images at different irradiation times. Each Raman image ($49 \times 53 \mu\text{m}^2$, 2500 pixels) was obtained within 6 s. Scale bar is $10 \mu\text{m}$. The Raman image is plotted using the Raman band (1340 cm^{-1}) of 4-NBT.

Comment 7: In Figure 6b-d what is the size of imaged area? How many seconds are needed to scan a higher area section (for example 0.5 mm^2)?

Response: Thanks for your comments. The size of imaged area in Figure 6b-d is $55 \times 66 \mu\text{m}^2$, and which was shown in *Methods* in the manuscript. We also added this information in the updated figure caption. For answering the above question from the reviewer, we have tried to scan a largest imaging area $600 \times 600 \mu\text{m}^2$ (0.36 mm^2 , 50×50 pixels) under the DuoScan mode (without moving the stage), which can be finished within 7 s. A typical example is shown in below Figure R12.

We have added it in the Supplementary Information (Supplementary Figure 14) and the corresponding discussion on page 19 as follows.

“...The enrichment of P-GERTs in the SLN was also confirmed by imaging a dissected lymph node (Supplementary Figure 14)...”

Figure R12. A wide-area Raman image (right, $600 \times 600 \mu\text{m}^2$, 2500 pixels) of a dissected lymph node (left) obtained within 7 s under the DuoScan mode. All scale bars are 100 μm . The Raman image is plotted using the Raman band (1340 cm^{-1}) of 4-NBT.

Comment 8: For *in vivo* imaging depth penetration is important. Did the authors perform any depth penetration study using the SERS tags? Lymph nodes are relative close to the skin; has any other deeper experiment been done?

Response: Thanks for reviewer's great comments. Yes, the Raman penetration is very important. Indeed, we performed a very preliminary test of the depth penetration measurements on optical phantom (agarose gels) using the P-GERTs. It has been found that the penetration depth in agarose gel is in the range of 2-4 cm when using the 785 nm laser (see below Figure R13) when the concentration of P-GERTs is 30 pM. Additional experiments show that the penetration depth decreases dramatically in the muscle and is only in the range of several millimeters. Further detailed experiments are still under way. Since this manuscript is only focusing on the improvement of the bioimaging speed using P-GERTs, we do not include these results and plan to publish them later.

Figure R13. Penetration depth of Raman signal with SERS tags in agarose gels.

Reviewers' comments:

Reviewer #1 (Remarks to the Author):

I think the authors did a fantastic job addressing all the many points highlighted by all the reviewers, including myself. The quality and clarity of the manuscript has been highly improved in this revised version and I think this manuscript can be now accepted for publication.

Reviewer #2 (Remarks to the Author):

The manuscript has been largely improved over the previous version, and this paper can be published in Nature Communications if the following two points are properly addressed.

1. There are several expressions that overly emphasized and exaggerated the results. All these should be toned down. For example, in Discussion (line 459-465), the following sentences "The nano-Raman measurements unquestionably confirm the single-NP sensitivity in solid state and additionally show ultra-high NP uniformity and super-long photostability, evaluated at a single-NP level. The huge SERS performance improvements of P-GERTs," The authors should remove 1) unquestionably, 2) ultra-high, 3) super-long, and 4) huge. The authors should also try to find and tone down other expressions like these.

2. It seems that the main advance here is bioimaging with Raman probes (high-speed, high-resolution single-cell Raman imaging and high-contrast wide-area in vivo SLN Raman imaging within a short period of time). Why all these are possible with the authors' probes and their method should be more clearly mentioned in the introduction and conclusion/discussion with scientific reasonings.

Reviewer #3 (Remarks to the Author):

In the present manuscript, some of the questions have been well clarified, however, there are still some of them unclear. In the revised version authors emphasized the fabrication of ultrabright gap-enhanced Raman tags, however when a 785 laser is used the Raman intensity is decreased. They noticed that they could constantly detect Raman signals from the MS 2G GERTs for relatively high concentrations, such as 10 pM, 1 pM, and 100 fM, but only signals in about a quarter of measurements for the concentration of 10 fM, and signals in very few cases for the concentration of 1 fM, which is commonly reported in the literature. Thus, the improvement of the particles is not so extraordinary. Moreover authors claim that their particles show slightly worse uniformity than the ones in ref: Nature Nanotechnology, 2011, 6, 452-460. Therefore the overall improvements are unclear.

The imaging speed is relative fast, but could it be related to the measurement set-up, data processing method...and not the brightness of the particles? Which is the minimum amount of particles that can be added per cell to perform these fast images? And resolution?

Authors did not show any experiments with alive cells and there are already studies with such experiments, for example Chem. Mater. 2016, 28, 6779–6790. Thus, I think is an important experiment to check as the stability of the particles can be strongly affected in real conditions.

Response to reviewers' comments

Reviewer 1

Comment 1: I think the authors did a fantastic job addressing all the many points highlighted by all the reviewers, including myself. The quality and clarity of the manuscript has been highly improved in this revised version and I think this manuscript can be now accepted for publication.

Response: Thanks so much for your recognition and putting so many efforts in reviewing our manuscript.

Reviewer 2

Comment 1: There are several expressions that overly emphasized and exaggerated the results. All these should be toned down. For example, in Discussion (line 459-465), the following sentences "The nano-Raman measurements unquestionably confirm the single-NP sensitivity in solid state and additionally show ultra-high NP uniformity and super-long photostability, evaluated at a single-NP level. The huge SERS performance improvements of P-GERTs," The authors should remove 1) unquestionably, 2) ultra-high, 3) super-long, and 4) huge. The authors should also try to find and tone down other expressions like these.

Response: Thank the reviewer for pointing out the statements overly emphasized in the previous version. **According to the reviewer's recommendation, we have found and removed the exaggerated words in the sentences as follows.**

Page 21-22: "The nano-Raman measurements **unquestionably** confirm the single-NP sensitivity in solid state and additionally show **ultra-high decent** NP uniformity and photostability, evaluated at a single-NP level. The **huge high** SERS **performance signal** improvements of P-GERTs are attributed to the larger surface area for molecular decoration, large Raman scattering cross-section of 4-NBT molecules and the difference in shell morphology, where S-GERTs are in a complete piece with a

smooth surface while P-GERTs have a petal-like structure with multiple nanogaps.”

Page 4: “In contrast, P-GERTs exhibit a continuous internal nanogap but a petal-like shell structure with a **rather** rough surface, both decorated with 4-NBT reporters (Fig. 1a i).”

Page 5: “The overall diameter of P-GERTs is 66 ± 4 nm with a **quite** monodispersed particle size and morphology (Fig. 1c).”

Page 10: “Fig. 2d indicates that the contribution of the Raman enhancement from the internal and external near-field hot spots (shaded area) becomes **dramatically** significant when the incubation time is longer than 2 min (namely, after forming petal-like shell structures) for both 633 nm and 785 nm lasers excitation (see more details in Supplementary Figure 6).”

Page 13: “The extinction and Raman spectra in aqueous solution of different pH values (from 5 to 10) and in 10% fetal bovine serum (FBS) and saline for 72 h show **quite** stable **intensity intensities** and unchanged spectral profiles (Supplementary Figure 10).”

Page 15: “More importantly, **super-long** single-NP photostability examinations of P-GERTs indicate that their SERS signals remained quite stable after continuous laser irradiation (110 μ W) for 1800 s (Fig. 4e).”

Comment 2: It seems that the main advance here is bioimaging with Raman probes (high-speed, high-resolution single-cell Raman imaging and high-contrast wide-area *in vivo* SLN Raman imaging within a short period of time). Why all these are possible with the authors' probes and their method should be more clearly mentioned in the introduction and conclusion/discussion with scientific reasonings.

Response: Thanks for reviewer’s comments. According to the reviewer’s suggestion,

we have added a clearer description to emphasize the importance/advantages of probe characteristics, imaging system and data processing method in the introduction and discussion part as follows.

Page 3, Introduction part: “... Due to the ultra-strong Raman signal of P-GERTs, ultra-fast laser scanning system (in a mode of moving the laser spot instead of the stage) and data processing method, high-speed and high-resolution cell imaging (2500 pixels for $55 \times 66 \mu\text{m}^2$ field of view) can be obtained within 6 s, one order of magnitude faster than before...”

Page 16, “...Such large Raman enhancement of P-GERTs down to a single-particle level may further allows us to improve the imaging speed with a further lower laser power, which is crucial to minimize the photodamage to biological samples. ...”

Page 21-22, Discussion part: “... Due to the ultra-strong Raman signal of P-GERTs, ultra-fast laser scanning technique and data processing method, we have achieved high-speed high-resolution single-cell Raman imaging (50×50 pixels for $55 \times 66 \mu\text{m}^2$ field of view) acquired within 6 s with a laser power $370 \mu\text{W}$ and acquisition time of 0.7 ms/pixel ...”

Reviewer #3

Comment 1: In the present manuscript, some of the questions have been well clarified, however, there are still some of them unclear. In the revised version authors emphasized the fabrication of ultrabright gap-enhanced Raman tags, however when a 785 laser is used the Raman intensity is decreased. They noticed that they could constantly detect Raman signals from the MS 2G GERTs for relatively high concentrations, such as 10 pM, 1 pM, and 100 fM, but only signals in about a quarter

of measurements for the concentration of 10 fM, and signals in very few cases for the concentration of 1 fM, which is commonly reported in the literature. Thus, the improvement of the particles is not so extraordinary. Moreover authors claim that their particles show slightly worse uniformity than the ones in ref: Nature Nanotechnology, 2011, 6, 452-460. Therefore, the overall improvements are unclear.

Response: Sorry for unclear description about the improvement of P-GERTs. We claimed previously in the manuscript: “We could constantly detect Raman signals from the MS P-GERTs for relatively high concentrations, such as 10 pM, 1 pM, and 100 fM, but only signals in about a quarter of measurements for the concentration of 10 fM, and signals in very few cases for the concentration of 1 fM”. In fact, this phenomenon is related to the number of nanoparticles in the laser spot instead of the Raman intensity of nanoparticles. When the average number of particles in the laser spot is less than one, this behavior may occur, regardless of the brightness of the single particles, as long as the brightness of the particle can reach the sensitivity of the single-particle detection. As we showed in Figure 3e, 10 fM and 1 fM P-GERT solutions correspond to only 0.2 and 0.02 particles per probing volume, respectively. Thus, we could only detect signals in about a quarter of measurements for the concentration of 10 fM and signals in very few cases for the concentration of 1 fM. We want to emphasize that this behavior occurs as long as it is single-particle detection in liquid state, regardless of the brightness of the particles.

Moreover, we carefully compared the SERS performance of P-GERTs and that of Au-NNPs, which were reported in Nature Nanotechnology, 2011, 6, 452-460. **First**, for AFM-correlated single-particle nano-Raman analysis, the measurements of P-GERTs in our work were performed in a more challenge condition: 633 nm laser, 10 s acquisition time, 100× objective lens, and 110 μW power. In contrast, the measurements of Au-NNPs were performed with 633 nm laser, 10 s acquisition time, and 100× objective lens, but the laser power was increased to 650 μW. **Second**, although the Raman EFs of two types of NPs are in a similar level (Au-NNPs: 1.0×10^8 to 5.0×10^9 ; P-GERTs: $\sim 5.4 \times 10^9$), the embedded Raman reporter molecules are ~ 100 per Au-NNPs and the embedded Raman molecules are ~ 6000 per P-GERTs.

Therefore, the overall Raman signal of single P-GERTs is much larger than that of Au-NNPs. This has been emphasized previously as one of the most crucial factors for high-speed imaging. In addition, the comparison of Raman signals of a large number of aqueous P-GERTs and Au-NNPs was performed. For Au-NNPs (see Figure 4 in Nature Nanotechnology, 2011, 6, 452-460), the Raman intensity is ~550 counts with 633 nm laser, 10 s exposure time, 300 μ W laser power and 0.5 nM concentration; and for P-GERTs (see Figure 1 in our manuscript), the Raman intensity is ~30000 counts with 633 nm laser, 2 s exposure time, 15 mW laser power and 0.2 nM concentration. After normalization of laser power, exposure time and particle concentration, the Raman intensity of P-GERTs (5000 counts/s·mW·nM) is more than one order larger than that of Au-NNPs (~370 counts/s·mW·nM). **Third**, Au-NNPs reported in Nature Nanotechnology (2011, 6, 452-460) need the anchored DNA strands as spacer to form the nanogaps. While the nanogap is directly formed using the embedded 4-NBT molecules for P-GERTs, and they are more cost-effective and easily prepared. In summary, P-GERTs, compared to Au-NNPs, show great improvements in overall SERS intensity, the ultra-high brightness allows for high-speed bioimaging with a low laser power and are more cost-effective and easy preparation for practical biomedical application.

Therefore, we have added the corresponding discussion in the revision as follows.

Page 6: "...More importantly, we found that the total Raman intensity of P-GERTs is more than two orders of magnitude higher than that of S-GERTs under the same experimental condition, estimated by comparing the Raman band at 1340 cm^{-1} for 4-NBT and at 1055 cm^{-1} for 1,4-BDT. It was also found that the Raman intensity of P-GERTs is roughly one order larger than that of the DNA-bridged SERS tags previously reported, after normalizing all experimental parameters (see more details in Supplementary Information). Different SERS performances can be explained by the finite-difference time-domain (FDTD) calculated electro-magnetic field distributions (insets in Fig. 1e)..."

Comment 2: The imaging speed is relative fast, but could it be related to the

measurement set-up, data processing method...and not the brightness of the particles? Which is the minimum amount of particles that can be added per cell to perform these fast images? And resolution?

Response: Thank the reviewer for raising this concern. Such high imaging speed can be attributed to the combination of the improvements in brightness of the particles, imaging system, and data processing method. **First and the most important,** P-GERTs with multiple electromagnetic hot spots can provide sufficient Raman signals, which greatly shorten the signal acquisition time. We can get Raman spectra with a good signal-to-noise ratio within ultrashort acquisition time of 0.7 ms per pixel. Please refer to the answer to the above **comment 1** of the **reviewer #3** for the brightness improvement compared to the previous nanoparticles with a similar structure. In addition, the step-by-step imaging mode, realizing rapid movement of the laser spot by the rotation of galvo mirrors, and data processing line-by-line in parallel in 'SWIFT' mode (Horiba) help time saving.

Therefore, we have added more discussion about the importance of the combination of probe characteristics, imaging system and data processing method in the introduction and discussion part as follows.

Page 3, Introduction part: "... Due to the ultra-strong Raman signal of P-GERTs, ultra-fast laser scanning system (in a mode of moving the laser spot instead of the stage) and data processing method, high-speed and high-resolution cell imaging (2500 pixels for $55 \times 66 \mu\text{m}^2$ field of view) can be obtained within 6 s, one order of magnitude faster than before..."

Page 21-22, Discussion part: "... Due to the ultra-strong Raman signal of P-GERTs, ultra-fast laser scanning technique and data processing method, we have achieved high-speed high-resolution single-cell Raman imaging (50×50 pixels for $55 \times 66 \mu\text{m}^2$ field of view) acquired within 6 s with a laser power $370 \mu\text{W}$ and acquisition time of 0.7 ms/pixel..."

Based on the results of single nanoparticle nano-Raman analysis, we estimated that the signal intensity of the single P-GERTs was about 10 counts under the condition of ultrafast cell imaging (0.7 ms, 3.6 mW). Then we selected cell Raman images with different signal intensities (strong, medium, and weak) and estimated the number of labelled P-GERTs. The cell Raman image (as shown in below Figure R1a) with excellent signals that can completely show the morphology of the cell contains about 5,000 particles. Cells Raman images (as shown in below Figure R1 b and c) with medium signals, which can basically show the outline of the cells, contains ~500 - 1000 particles. The cell Raman image (as shown in below Figure R1d) with weak signals that can only show a part of the cell contains ~80 particles. The resolutions of all cell images are ~2500 pixels.

Therefore, we have added below Figure R1 into the Supplementary Information and added more discussion about the minimum number of nanoparticles for the cell labeling as follows.

Page 17: "... These result in the possibility to detect a single cell labelled with less than 100 P-GERTs based on the aforementioned single-NP AFM nano-Raman measurements (refer to Supplementary Figure 12 for more details) ..."

Figure R1. Representative SERS imaging of cells containing different numbers of P-GERTs: bright-field images (left) and Raman images of different signal intensities labelled with (a) ~5000, (b) ~1000, (c) ~450 and (d) ~80 P-GERTs. The resolution of all cell images is ~2500 pixels. Scale bar is 10 μm . The Raman image is plotted using the Raman band (1340 cm^{-1}) of 4-NBT.

Comment 3: Authors did not show any experiments with alive cells and there are already studies with such experiments, for example Chem. Mater. 2016, 28, 6779–6790. Thus, I think is an important experiment to check as the stability of the particles can be strongly affected in real conditions.

Response: Thanks for the reviewer's comments. According to the reviewer's suggestion, we have performed time-lapse live cell SERS imaging to study the stability of P-GERTs in real conditions. After cells were adhered to quartz plates and incubated with P-GERTs (0.05 nM) for 6 h, the cells were transferred to a live cell incubator (Stage Top Incubator, Okolab, Italy) for further culture and SERS imaging. The Okolab incubator can fit well in the XY stage of the microscope, connecting to a control system of temperature, gas and humidity to create the proper environment for live cell imaging right on the microscope stage. Time-lapse live cells SERS imaging were performed to evaluate the stability of P-GERTs in live cells in an inverted

Raman system (Horiba, Xplora INV). During the living cell imaging, the acquisition time per pixel was increased to 1 ms and then the single live-cell imaging (2500 pixels for $80 \times 32 \mu\text{m}^2$ field of view) was obtained within 12 s. We continuously recorded live multiple-cell images (2500 pixels for $135 \times 180 \mu\text{m}^2$ field of view) for a total of 30 min (shown in below Figure R2 a) and single-cell images for a total of 30 min (shown in below Figure R2 c) both with a time interval of 5 min. Under a low magnification, multiple-cell images show almost identical color patterns, meaning that P-GERTs maintain a sufficient stability in live cells during the imaging process. This can be also confirmed by the three representative Raman spectra at the point 1 (indicated in panel a) before, in the middle, and after the imaging process (Figure R2 b). In contrast, the live single-cell images with a higher magnification may offer more information. For example, P-GERTs are more easily diffused or aggregated in a location indicated by a pink arrow than that indicated by a white arrow (Figure R2 c). These behaviors may be potentially linked to the biological event of cells. So far this is only our preliminary work and we show it here as a demonstration of our P-GERTs for the possibility of live-cell SERS high-speed imaging. We will perform more detailed investigation in future work.

Furthermore, we compared our live cell imaging with the previous work (Chem. Mater. 2016, 28, 6779–6790). In the previous work, the live cell imaging was performed with Au nanostars and Raman signals were acquired with 1 s acquisition time per pixel at 15 mW laser power. In our experiment, the ultra-high brightness of P-GERTs allows for high-speed live cell imaging (1 ms acquisition time per pixel) with a low laser power (0.37 mW). This causes much less photo-damage to cells and is more favorable for practical live cell tracking.

Therefore, we have added below Figure R2 into the Supplementary Information and added the corresponding discussion on page 18 as follows.

Page 18: “...We additionally demonstrated that P-GERTs exhibited excellent long-term photostability during high-speed 30-min continuous cell imaging process (Supplementary Figure 14) and the great potential for high-speed live-cell imaging

(Supplementary Figure 15)....”

Figure R2. Time-lapse live cell Raman images obtained from (a) multiple cells ($135 \times 180 \mu\text{m}^2$, 2500 pixels) and (c) a single cell ($80 \times 32 \mu\text{m}^2$, 2500 pixels). (b) Three representative Raman spectra at the point 1 (indicated in panel a) before, in the middle, and after the 30-min imaging process. Scale bar is 50 and 20 μm in panel a and c, respectively. All Raman image are plotted using the Raman band (1340 cm^{-1}) of 4-NBT.

REVIEWERS' COMMENTS:

Reviewer #2 (Remarks to the Author):

The authors addressed the reviewers' comments well, and the manuscript has been revised accordingly. This reviewer now recommends publication of this work in Nature Communications.

Reviewer #3 (Remarks to the Author):

In this version authors have addressed correctly the questions and recommendations that reviewers proposed. Therefore, the quality of the manuscript has improved considerably and I think that it can be accepted. However, I still think that the bibliography should be revised and some references related to SERS tags such as (<https://doi.org/10.1002/cnma.201500221>) and the ones mentioned during the corrections such as Chem. Mater. 2016, 28, 6779–6790 should be included in the final version.

Response to reviewers' comments

Reviewer #2

Comment: The authors addressed the reviewers' comments well, and the manuscript has been revised accordingly. This reviewer now recommends publication of this work in Nature Communications.

Response: Thanks so much for your recognition and putting so many efforts in reviewing our manuscript.

Reviewer #3

Comment: In this version authors have addressed correctly the questions and recommendations that reviewers proposed. Therefore, the quality of the manuscript has improved considerably and I think that it can be accepted. However, I still think that the bibliography should be revised and some references related to SERS tags such as (<https://doi.org/10.1002/cnma.201500221>) and the ones mentioned during the corrections such as Chem. Mater. 2016, 28, 6779–6790 should be included in the final version.

Response: Thanks for suggestion of these relevant excellent papers. **We have added these papers as references 10 and 18 to the list of references.**